# Lookahead Sharpness-Aware Minimization

## Abstract

Sharpness-Aware Minimization (SAM), which performs gradient descent on adversarially perturbed weights, can improve generalization by identifying flatter minima. However, recent studies have shown that SAM may suffer from convergence instability and oscillate around saddle points, resulting in slow convergence and inferior performance. To address this problem, we propose the use of a lookahead mechanism in the methods of extra-gradient and optimistic gradient. By examining the nature of SAM, we simplify the extrapolation procedure, resulting in a more efficient algorithm. Theoretical results show that the proposed method converge to a stationary point and escape saddle points faster. Experiments on standard benchmark datasets also verify that the proposed method outperforms the SOTAs, and converge more effectively to flat minima.

## 1 Introduction

Deep learning models have been successful in various real-world applications (LeCun et al., 2015). However, highly over-parameterized neural networks may suffer from model overfitting and poor generalization (Zhang et al., 2021). Hence, reducing the performance gap between training and testing is an important research topic (Neyshabur et al., 2017). Recently, there have been a number of works exploring the close relationship between loss geometry and generalization performance. In particular, it has been observed that flat minima often imply better generalization (Chatterji et al., 2020; Jiang et al., 2020; Chaudhari et al., 2019; Dziugaite & Roy, 2017; Petzka et al., 2021).

To locate flat minima, a popular approach is based on Sharpness-Aware Minimization (SAM) (Foret et al., 2021). Recently, a number of variants have also been proposed (Kwon et al., 2021; Zhuang et al., 2022; Du et al., 2022b;a; Jiang et al., 2023; Liu et al., 2022). The main idea of SAM is to first add a (adversarial) perturbation to the weights and then perform gradient descent there. However, these methods are myopic as they only update their parameters based on the gradient of the adversarially perturbed parameters. Consequently, the model may converge slowly as it lacks good information about the loss landscape. Recent research has found that SAM can suffer from convergence instability near a saddle point (Kim et al., 2023; Compagnoni et al., 2023; Kaddour et al., 2022).

To alleviate the myopic problem, one possibility is to encourage the model to gather more information about the landscape by looking further ahead, and thus find better trajectory to converge (Leng et al., 2018; Wang et al., 2022). In game theory, two popular methods that can encourage the agent to look ahead are the method of extra-gradient (Korpelevich, 1976; Gidel et al., 2019; Lee et al., 2021) and its approximate cousin, the method of optimistic gradient (Popov, 1980; Gidel et al., 2019; Daskalakis & Panageas, 2018; Daskalakis et al., 2018; Mokhtari et al., 2020). Their key idea is to first perform an extrapolation step that looks one step ahead into the future, and then perform gradient descent based on the extrapolation result (Bohm et al., 2022). Besides game theory, similar ideas have also been proven successful in deep learning optimization (Zhou et al., 2021; Zhang et al., 2019; Lin et al., 2020a), and reinforcement learning (Liu et al., 2023). As SAM is formulated as a minimax optimization problem (Foret et al., 2021), this inspires us to also leverage an extrapolation step for better convergence.

In this paper, we introduce the look-ahead mechanism to SAM. Our main contributions are fourfold:

   (i) We incorporate the idea of extrapolation into SAM so that the model can gain more information about the landscape, and thus help convergence.

   (ii) By studying the SAM updates, we propose to reduce the computational cost by removing some steps from a straightforward application of extra-gradient or optimistic gradient ascent.

(iii) We provide theoretical guarantees that they converge to stationary points at the same rate as SAM.

(iv) Experimental results show that the proposed method has better performance, and also converge to a flatter minimum.

## 2 BACKGROUND

**Sharpness-Aware Minimization (SAM).** SAM (Foret et al., 2021) attempts to improve generalization by finding flat minima (Foret et al., 2021; Wen et al., 2023). This is achieved by minimizing the worst-case loss within some perturbation radius. Mathematically, it is formulated as the following minimax optimization problem (Jiang et al., 2023; Liu et al., 2022; Zhao, 2022; Du et al., 2022a):

$$\min_{\boldsymbol{w}\in\mathbb{R}^n} \max_{\boldsymbol{\epsilon}:\|\boldsymbol{\epsilon}\|\leq\rho} L(\boldsymbol{w}+\boldsymbol{\epsilon}), \tag{1}$$

where $L$ is the loss function, $\boldsymbol{w}$ is the model parameter, and $\boldsymbol{\epsilon}$ is the perturbation whose magnitude is bounded by $\rho$. By taking first-order approximation on the objective, the optimal $\boldsymbol{\epsilon}$ for the maximization subproblem can be obtained as (Foret et al., 2021; Kwon et al., 2021)

$$\boldsymbol{\epsilon}^*(\boldsymbol{w}) = \frac{\rho\nabla_{\boldsymbol{w}}L(\boldsymbol{w})}{\|\nabla_{\boldsymbol{w}}L(\boldsymbol{w})\|}. \tag{2}$$

Thus, (1) can be solved by performing gradient descent

$$\boldsymbol{w}_t = \boldsymbol{w}_{t-1} - \eta\nabla_{\boldsymbol{w}_{t-1}}L\left(\boldsymbol{w}_{t-1} + \boldsymbol{\epsilon}_{t-1}\Big|_{\boldsymbol{\epsilon}_{t-1}=\frac{\rho\nabla_{\boldsymbol{w}_{t-1}}L(\boldsymbol{w}_{t-1})}{\|\nabla_{\boldsymbol{w}_{t-1}}L(\boldsymbol{w}_{t-1})\|}}\right) \tag{3}$$

at iteration $t$, where $\eta$ is the learning rate.

As SAM requires two forward-backward calculations in each iteration, computationally it is more expensive than standard empirical risk minimization (ERM). Recently, a number of variants (including AE-SAM (Jiang et al., 2023), LookSAM (Liu et al., 2022), SS-SAM (Zhao, 2022), ESAM (Du et al., 2022a)) have been proposed to reduce this cost by using SAM only in iterations that it is likely to be useful, and use ERM otherwise. For example, Jiang et al. (2023) proposes the AE-SAM, which uses SAM only when the loss landscape is locally sharp, with sharpness being approximated efficiently by $\|\nabla L(\boldsymbol{w}_t)\|^2$. It is shown that $\|\nabla L(\boldsymbol{w}_t)\|^2$ can be modeled empirically with a normal distribution $\mathcal{N}(\mu_t, \sigma_t^2)$, in which $\mu_t$ and $\sigma_t^2$ are estimated in an online manner by exponential moving average as:

$$\mu_t = \delta\mu_{t-1} + (1-\delta)\|\nabla L(\boldsymbol{w}_t)\|^2, \quad \sigma_t^2 = \delta\sigma_{t-1}^2 + (1-\delta)(\|\nabla L(\boldsymbol{w}_t)\|^2 - \mu_t)^2, \tag{4}$$

where $\delta \in (0,1)$ controls the forgetting rate. When the loss landscape is locally sharp (i.e., $\|\nabla L(\boldsymbol{w}_t)\|^2 \geq \mu_t + c_t\sigma_t$), SAM is used; otherwise, ERM is used. The threshold $c_t$ is decreased linearly from $\kappa_2$ to $\kappa_1$ by the schedule:

$$c_t = \frac{t}{T}\kappa_1 + \left(1 - \frac{t}{T}\right)\kappa_2, \tag{5}$$

where $T$ is the total number of iterations.

Besides reducing the training cost, another direction is to improve the performance. For example, ASAM (Kwon et al., 2021) introduces adaptive sharpness to improve generalization; and GSAM (Zhuang et al., 2022) uses a new surrogate loss function that focuses more on sharpness.

**Extra-Gradient (EG) (Korpelevich, 1976).** Consider the minimax problem:

$$\min_{x\in\mathbb{R}^m} \max_{y\in\mathbb{R}^n} f(x,y). \tag{6}$$

EG performs gradient descent-ascent (GDA), i.e., gradient ascent $\nabla_y f(x,y)$ on $y$ and gradient descent $-\nabla_x f(x,y)$ on $x$. Specifically, let $z := [x,y]^\top$ and $F(z) := [\nabla_x f(x,y), -\nabla_y f(x,y)]^\top$. At the $t$th iteration, the EG update can be written as:

$$\bar{z}_t = z_t - \eta F(z_t), \quad z_{t+1} = z_t - \eta F(\bar{z}_t), \tag{7}$$

where $\eta$ is the learning rate. While the update on $z_{t+1}$ is the usual descent, the update on $\bar{z}_t$ performs an extra extrapolation step which avoids shortsightedness of both players ($x$ and $y$) by looking one

step ahead into the future (Gidel et al., 2019; Bohm et al., 2022; Jelassi et al., 2020; Pethick et al., 2022). EG has been widely used in two-player zero-sum games (Fudenberg & Tirole, 1991). In machine learning, this has been used in the training of generative adversarial network (GAN) (Gidel et al., 2019), poker games (Lee et al., 2021), and more recently, faster optimization in large-batch training (Lin et al., 2020a; Xu et al., 2019).

As shown in (7), each EG iteration requires computing the gradients w.r.t. $x$ and $y$ twice. To reduce the cost, the method of optimistic gradient (OG) (Popov, 1980) stores the past gradient $F(\overline{z}_{t-1})$ and reuses it in the next extrapolation step. The update in $\overline{z}_t$ is thus changed to:

$$\overline{z}_t = z_t - \eta F(\overline{z}_{t-1}). \tag{8}$$

Hence, the gradients w.r.t. $x$ and $y$ only need to be computed once in each iteration. It can be shown that OG enjoys a similar convergence rate as EG (Gidel et al., 2019), and has been commonly used in solving differentiable minimax games (Gidel et al., 2019; Liang & Stokes, 2019; Daskalakis & Panageas, 2018; Daskalakis et al., 2018).

## 3 LOOKAHEAD IN SAM

While SAM has shown improved generalization on various tasks, recently it is observed that SAM can have convergence instability near a saddle point (Kim et al., 2023; Compagnoni et al., 2023), leading to slow convergence and poor performance. As an example, consider minimizing the following quadratic function as in (Compagnoni et al., 2023): $\min_{x \in \mathbb{R}^2} \ell(x) \equiv x^\top H x$, where $H \equiv \text{diag}(-1, 1)$. The saddle point is at $[0, 0]$. we run SAM with an initial $x_0 = [0.02, 0.02]$, and SGD optimizer with learning rate of 0.005. In every SGD step $t$, we add Gaussian noise $\epsilon'_t \sim \mathcal{N}(0, 0.01)$ to the gradient as in (Compagnoni et al., 2023). As can be seen from Figure 1, SAM is trapped in the saddle point.

In the following, inspired by the method of extra-gradient (EG), we propose a number of lookahead mechanisms to alleviate the convergence problem of SAM.

### 3.1 SAM+EG

A straightforward solution is to use EG on SAM's minimax optimization problem in (1). This leads to the following update at iteration $t$:

$$\hat{\boldsymbol{w}}_t = \boldsymbol{w}_{t-1} - \eta_t \nabla_{\boldsymbol{w}_{t-1}} L(\boldsymbol{w}_{t-1} + \boldsymbol{\epsilon}_{t-1}), \tag{9}$$
$$\hat{\boldsymbol{\epsilon}}_t = \Pi(\boldsymbol{\epsilon}_{t-1} + \eta'_t \nabla_{\boldsymbol{\epsilon}_{t-1}} L(\boldsymbol{w}_{t-1} + \boldsymbol{\epsilon}_{t-1})), \tag{10}$$
$$\boldsymbol{w}_t = \boldsymbol{w}_{t-1} - \eta_t \nabla_{\hat{\boldsymbol{w}}_t} L(\hat{\boldsymbol{w}}_t + \hat{\boldsymbol{\epsilon}}_t), \tag{11}$$
$$\boldsymbol{\epsilon}_t = \Pi(\boldsymbol{\epsilon}_{t-1} + \eta'_t \nabla_{\hat{\boldsymbol{\epsilon}}_t} L(\hat{\boldsymbol{w}}_t + \hat{\boldsymbol{\epsilon}}_t)), \tag{12}$$

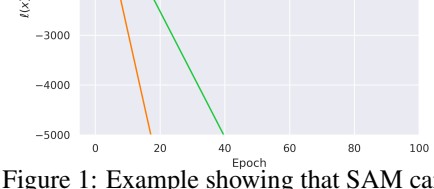

Figure 1: Example showing that SAM can be trapped in a saddle point.

where $\Pi(\cdot)$ in (10) is the projection $\Pi(\boldsymbol{\epsilon}) := \arg\min_{\boldsymbol{\epsilon}':\|\boldsymbol{\epsilon}'\| \leq \rho} \|\boldsymbol{\epsilon} - \boldsymbol{\epsilon}'\| = \frac{\boldsymbol{\epsilon}}{\max(1, \|\boldsymbol{\epsilon}\|/\rho)}$, $\eta'_t$ is the learning rate. The learning rates in (10) and (12) are set to 1, as is commonly used in SAM (Foret et al., 2021) and its variants. As can be seen from (9)-(12), each update requires four gradient computations, which can be expensive. Moreover, gradient descent ascent (GDA) on (1), as is performed in EG, converges at a rate of $O(T^{-\frac{1}{4}})$ for non-convex strongly-concave settings (Lin et al., 2020b; Mahdavinia et al., 2022) (might be even slower in general non-convex non-concave settings), where $T$ is the total number of epochs. This is much slower than the $O(1/\sqrt{T})$ rate of SAM (Andriushchenko & Flammarion, 2022) [1].

### 3.2 SAM+OG

In each epoch, SAM+EG has to compute the gradient w.r.t $\boldsymbol{w}$ and $\boldsymbol{\epsilon}$ twice in each iteration, which can be expensive when used on large deep networks. Motivated by the optimistic gradient (OG) approach (Korpelevich, 1976), at iteration $t$, we rewrite (9) as:

$$\hat{\boldsymbol{w}}_t = \boldsymbol{w}_{t-1} - \eta_t \nabla_{\hat{\boldsymbol{w}}_{t-1}} L(\hat{\boldsymbol{w}}_{t-1} + \hat{\boldsymbol{\epsilon}}_{t-1}),$$

and (10) as

$$\boldsymbol{\epsilon}_t = \Pi(\boldsymbol{\epsilon}_{t-1} + \eta'_t \nabla_{\hat{\boldsymbol{\epsilon}}_{t-1}} L(\hat{\boldsymbol{w}}_{t-1} + \hat{\boldsymbol{\epsilon}}_{t-1})).$$

---

[1]Note that though the rate of SAM is faster than EG, SAM (as well as our proposed method) may not converge to the exact stationary point of (1) (the definition of stationary point will be introduced in Sec. A.5).

Since $\nabla_{\hat{w}_{t-1}} L(\hat{w}_{t-1} + \hat{\epsilon}_{t-1})$ and $\nabla_{\hat{\epsilon}_{t-1}} L(\hat{w}_{t-1} + \hat{\epsilon}_{t-1})$ has already been computed in $t-1$, we only need to compute the gradient w.r.t. $w$ and $\epsilon$ once. However, SAM+OG still belongs to GDA-based method. Similar to SAM+EG, it converges at a rate of $O\left(T^{-\frac{1}{4}}\right)$ (Mahdavinia et al., 2022), which is also much slower than the rate of SAM.

### 3.3 SAM+EG WITH APPROXIMATED MAX ORACLE

As the maximization w.r.t. $\epsilon$ in (1) has an approximated solution (2) (approximated max oracle) following (Foret et al., 2021), one can directly apply extra-gradient (Leng et al., 2018) on the minimization problem in (3), which is easier than the original minimax optimization formulation. The EG update is then:

$$\hat{w}_t = w_{t-1} - \eta_t \nabla_{w_{t-1}} L(w_{t-1} + \hat{\epsilon}_t),$$

$$w_t = w_{t-1} - \eta_t \nabla_{\hat{w}_t} L(\hat{w}_t + \epsilon'_t),$$

(13)

where $\hat{\epsilon}_t \equiv \frac{\rho \nabla_{w_{t-1}} L(w_{t-1})}{\|\nabla_{w_{t-1}} L(w_{t-1})\|}$ and $\epsilon'_t \equiv \frac{\rho \nabla_{\hat{w}_t} L(\hat{w}_t)}{\|\nabla_{\hat{w}_t} L(\hat{w}_t)\|}$. Following SAM, we drop $\nabla_{w_t} \hat{\epsilon}_t$ and $\nabla_{w_t} \epsilon'_t$ to make the gradient computation feasible for large neural networks. This method is named SAM+EG with Approximated Max Oracle (SAM+EG-AMO). Comparing with SAM, it contains two approximated gradient steps, and the accumulation of errors leads to bad performance. As will be empirically shown in Section 5.1, this method does not work well.

### 3.4 EG-SAM, OG-SAM, AND AO-SAM

To improve the convergence rate for SAM+EG, and achieve better performance than SAM+EG-AMO, we propose the following method that incorporates both the lookahead and the approximated solution in (2):

$$\hat{w}_t = w_{t-1} - \eta_t \nabla_{w_{t-1}} L(w_{t-1}),$$

(14)

$$\hat{\epsilon}_t = \frac{\rho \nabla_{w_{t-1}} L(w_{t-1})}{\|\nabla_{w_{t-1}} L(w_{t-1})\|},$$

(15)

$$w_t = w_{t-1} - \eta_t \nabla_{\hat{w}_t} L(\hat{w}_t + \hat{\epsilon}_t).$$

(16)

The updates in (14)-(16) can be interpreted as that we first perform a lookahead step (14) at $w_{t-1}$, add perturbation $\hat{\epsilon}_t$ in (15) as in SAM, and then update $w_{t-1}$ using the information on $L(\hat{w}_t + \hat{\epsilon}_t)$ via (16). In practice, stochastic gradients are used instead of batch gradients in the update. Let the loss on the $i$th sample be $\ell_i(w_t)$. The whole procedure, which will be called Extra-Gradient SAM (EG-SAM), is shown in Algorithm 1.

---

**Algorithm 1:** Extra-Gradient SAM (EG-SAM).

**Input:** Training set $S$, number of epochs $T$, batch size $b$, learning rate $\eta_0$, $\rho_0$, $w_0$.

1 **for** $t = 1, 2, \ldots, T$ **do**
2    sample a minibatch $I_t$ from $S$ with size $b$;
3    $\hat{w}_t = w_{t-1} - \eta_t \nabla_{w_{t-1}} \left[\frac{1}{b} \sum_{i \in I_t} \ell_i(w_{t-1})\right]$;
4    $\hat{\epsilon}_t = \frac{\rho_t \nabla_{w_{t-1}} \frac{1}{b} \sum_{i \in I_t} \ell_i(w_{t-1})}{\|\nabla_{w_{t-1}} \frac{1}{b} \sum_{i \in I_t} \ell_i(w_{t-1})\|}$;
5    $w_t = w_{t-1} - \eta_t \nabla_{\hat{w}_t} \left[\frac{1}{b} \sum_{i \in I_t} \ell_i(\hat{w}_t + \hat{\epsilon}_t)\right]$;
6 **return** $w_T$.

---

**Algorithm 2:** Optimistic-Gradient SAM (OG-SAM).

**Input:** Training set $S$, number of epochs $T$, batch size $b$, learning rate $\eta_0$, $\epsilon_0 = 0$, $w_0 = \mathbf{0}$, $\mathbf{g}_0 = \mathbf{0}$.

1 **for** $t = 1, 2, \ldots, T$ **do**
2    sample a minibatch $I_t$ from $S$ with size $b$;
3    $\hat{w}_t = w_{t-1} - \eta_t \mathbf{g}_{t-1}$;
4    $\hat{\epsilon}_t = \frac{\rho_t \nabla_{w_{t-1}} \frac{1}{b} \sum_{i \in I_t} \ell_i(w_{t-1})}{\|\nabla_{w_{t-1}} \frac{1}{b} \sum_{i \in I_t} \ell_i(w_{t-1})\|}$;
5    $\mathbf{g}_t = \nabla_{\hat{w}_t} \left[\frac{1}{b} \sum_{i \in I_t} \ell_i(\hat{w}_t + \hat{\epsilon}_t)\right]$;
6    $w_t = w_{t-1} - \eta_t \mathbf{g}_t$;
7 **return** $w_T$.

---

Similar to EG+SAM, EG-SAM has to compute the gradient w.r.t $w$ twice in each iteration, which can be expensive for large deep networks. Following the same idea in Sec. 3.2, at iteration $t$ we use OG and rewrite (14) as:

$$\hat{w}_t = w_{t-1} - \eta_t \nabla_{\hat{w}_{t-1}} L(\hat{w}_{t-1} + \hat{\epsilon}_{t-1}).$$

(17)

Since $\nabla_{\hat{\boldsymbol{w}}_{t-1}} L(\hat{\boldsymbol{w}}_{t-1} + \hat{\boldsymbol{\epsilon}}_{t-1})$ has already been computed in $t-1$, we only require to compute the gradient once. Equation (17) can be interpreted as the optimistic mirror descent method with $L_2$ norm (Chiang et al., 2012; Wei et al., 2021), i.e.,

$$\hat{\boldsymbol{w}}_t = \arg\min_{\boldsymbol{w}} \left\{ \eta_t \left\langle \boldsymbol{w}, \nabla_{\hat{\boldsymbol{w}}_{t-1}} L(\hat{\boldsymbol{w}}_{t-1} + \hat{\boldsymbol{\epsilon}}_{t-1}) \right\rangle + \frac{1}{2} \|\boldsymbol{w} - \boldsymbol{w}_{t-1}\|_2^2 \right\} = \boldsymbol{w}_{t-1} - \eta_t \nabla_{\hat{\boldsymbol{w}}_{t-1}} L(\hat{\boldsymbol{w}}_{t-1} + \hat{\boldsymbol{\epsilon}}_{t-1}),$$

which improves the performance by leveraging the information from the past gradient (Rakhlin & Sridharan, 2013), and has been widely used in online learning (Chiang et al., 2012) and game theory (Wei et al., 2021). The procedure, which will be called optimistic-gradient SAM (OG-SAM), is shown in Algorithm 2. Note that again we use the stochastic gradient which is more feasible in practice. Comparing with SAM+EG-AMO, OG-SAM inherits from SAM+OG, which guarantees convergence without accessing $\nabla_{\hat{\boldsymbol{w}}} \hat{\boldsymbol{\epsilon}}$ (Gidel et al., 2019). Therefore, though OG-SAM also does not compute $\nabla_{\hat{\boldsymbol{w}}} \hat{\boldsymbol{\epsilon}}$ in the updated scheme, it can achieve good performance.

OG-SAM still has to compute the gradient in each iteration, which can be expensive for large neural network. To alleviate this issue, we integrate AE-SAM's adaptive policy (Jiang et al., 2023) with OG-SAM. Assume that $\|\frac{1}{b}\sum_{i \in I_t} \nabla_{\boldsymbol{w}_t} \ell_i(\boldsymbol{w}_t)\|^2$ follows the normal distribution $\mathcal{N}(\mu_t, \sigma_t^2)$ with mean $\mu_t$ and variance $\sigma_t^2$. If $\|\frac{1}{b}\sum_{i \in I_t} \nabla_{\boldsymbol{w}_t} \ell_i(\boldsymbol{w}_t)\|^2$ is large (i.e., $\geq \mu_t + c_t \sigma_t$, where $c_t$ is varied as in (5)), we use OG-SAM. Otherwise, SGD (i.e., ERM) is used instead.

Recall that in (17), we need to access $\nabla_{\hat{\boldsymbol{w}}_{t-1}} L(\hat{\boldsymbol{w}}_{t-1} + \hat{\boldsymbol{\epsilon}}_{t-1})$ at iteration $t$. If $\|\frac{1}{b}\sum_{i \in I_{t-1}} \nabla_{\boldsymbol{w}_{t-1}} \ell_i(\boldsymbol{w}_{t-1})\|^2 < \mu_{t-1} + c_{t-1}\sigma_{t-1}$ in iteration $t-1$, SAM is not used and $\nabla_{\hat{\boldsymbol{w}}_{t-1}} L(\hat{\boldsymbol{w}}_{t-1} + \hat{\boldsymbol{\epsilon}}_{t-1})$ is not computed. In that case, we replace $\nabla_{\boldsymbol{w}_{t-1}} L(\hat{\boldsymbol{w}}_{t-1} + \hat{\boldsymbol{\epsilon}}_{t-1})$ with $\nabla_{\boldsymbol{w}_{t-1}} L(\boldsymbol{w}_{t-1})$. The whole procedure, which will be called Adaptive Optimistic Gradient SAM (AO-SAM), is shown in Algorithm 3.

---

**Algorithm 3:** Adaptive Optimistic Gradient SAM (AO-SAM).

---

**Input:** Training set $S$, number of epochs $T$, batch size $b$, learning rate $\eta$, $\boldsymbol{w}_0$, $\epsilon_0 = 0$, $\hat{\boldsymbol{w}}_0$, $\mu_0 = 0$, and $\sigma_0 = e^{-10}$.

1 **for** $t = 1, 2, \ldots, T$ **do**
2      sample a minibatch $I_t$ from $S$ with size $b$;
3      $\boldsymbol{g}_t = \frac{1}{b}\sum_{i \in I_t} \nabla_{\boldsymbol{w}_t} \ell_i(\boldsymbol{w}_t)$;
4      update $\mu_t$ and $\sigma_t$ as in AE-SAM (4);
5      **if** $\|\frac{1}{b}\sum_{i \in I_t} \nabla_{\boldsymbol{w}_t} \ell_i(\boldsymbol{w}_t)\|^2 \geq \mu_t + c_t \sigma_t$ **then**
6          $\hat{\boldsymbol{w}}_t = \boldsymbol{w}_{t-1} - \eta_t \boldsymbol{g}_{t-1}$ ;
7          $\hat{\boldsymbol{\epsilon}}_t = \frac{\rho_t \nabla_{\boldsymbol{w}_{t-1}} \frac{1}{b}\sum_{i \in I_t} \ell_i(\boldsymbol{w}_{t-1})}{\|\nabla_{\boldsymbol{w}_{t-1}} \frac{1}{b}\sum_{i \in I_t} \ell_i(\boldsymbol{w}_{t-1})\|}$;
8          $\boldsymbol{g}_t = \nabla_{\hat{\boldsymbol{w}}_t} \frac{1}{b}\sum_{i \in I_t} \ell_i(\hat{\boldsymbol{w}}_t + \hat{\boldsymbol{\epsilon}}_t)$;
9      $\boldsymbol{w}_t = \boldsymbol{w}_{t-1} - \eta_t \boldsymbol{g}_t$;
10 **return** $\boldsymbol{w}_T$.

---

## 4 ANALYSIS

### 4.1 EG-SAM'S ODE ON QUADRATIC LOSS

Consider the objective

$$\min_{\boldsymbol{w}} \ell(\boldsymbol{w}) \equiv \boldsymbol{w}^\top H \boldsymbol{w}. \tag{18}$$

Recall that the Ordinary Differential Equation (ODE) of SAM follows (Compagnoni et al., 2023):

$$d\boldsymbol{w}_\tau = -H \left( \boldsymbol{w}_\tau + \frac{\rho H \boldsymbol{w}_\tau}{\|H \boldsymbol{w}_\tau\|} \right) d\tau, \tag{19}$$

where $\tau$ is the time. For EG-SAM, note that its update in (14)-(16) can be rewritten as

$$d\boldsymbol{w}_\tau = -H \left( \boldsymbol{w}_\tau + \frac{\rho H \boldsymbol{w}_\tau}{\|H \boldsymbol{w}_\tau\|} - \eta_\tau H \boldsymbol{w}_\tau \right) d\tau. \tag{20}$$

where $\eta_\tau : \mathbb{R}^+ \to \mathbb{R}^+$ can be any function satisfying for every $\tau \in \mathbb{R}^+$ and $\epsilon' \in \mathbb{R}^+$, the following condition $\eta_\tau \geq \eta_{\tau+\epsilon'} > 0$ holds. The relationship between EG-SAM/SAM's ODE and their original updated scheme is in Appendix A.1.

Let $\{\lambda_i\}_{i=1}^d$ be the eigenvalues of $H$, and $\lambda_{\max}$ and $\lambda_{\min}$ be the maximum and minimum eigenvalues, respectively. For a given point $\boldsymbol{w}'$, define its region of attraction (ROA) (Chang et al., 2019) as the set such that all trajectories starting inside it converge to $\boldsymbol{w}'$ (Mao, 2007). Smaller ROA means less possibility to converge to that point.

The following Proposition shows that EG-SAM has a smaller ROA (i.e., less possibility) than SAM for any non-degenerate saddle point [2]. Proofs are in Appendix A.

**Proposition 4.1.** *For a non-degenerated saddle point, EG-SAM has a smaller ROA than SAM.*

This Proposition also indicates that SAM is easier to be trapped in a saddle point than EG-SAM. Moreover, the following shows that for the stationary point with smaller $\lambda_{\max}$, EG-SAM has larger ROA.

**Proposition 4.2.** *For non-degenerated stationary points $\boldsymbol{w}_1^*$ and $\boldsymbol{w}_2^*$, with the largest Hessian eigenvalues $\lambda_{\max}^1$ and $\lambda_{\max}^2$, respectively. If $\lambda_{\max}^1 > \lambda_{\max}^2$, then $\boldsymbol{w}_1^*$ has a smaller ROA than $\boldsymbol{w}_2^*$.*

As a smaller $\lambda_{\max}$ indicates a flatter minimum (Jastrzebski et al., 2020; Dinh et al., 2017; Kaur et al., 2022), this Proposition shows that a smaller $\lambda_{\max}$ implies a larger ROA for EG-SAM. Thus, EG-SAM has more chance to converge at flatter minima.

## 4.2 CONVERGENCE ANALYSIS

In this section, we study the convergence properties of EG-SAM, OG-SAM, and AO-SAM. Note that our analysis is different from those in the literature on extra-gradient (EG) (Gidel et al., 2019; Bohm et al., 2022; Jelassi et al., 2020; Pethick et al., 2022; Gorbunov et al., 2022; Cai et al., 2022) and optimistic gradient (OG) (Gidel et al., 2019; Liang & Stokes, 2019; Daskalakis & Panageas, 2018; Daskalakis et al., 2018; Mahdavinia et al., 2022). EG and OG assume that $f$ in (6) is (strongly) convex w.r.t. $x$, and (strongly) concave, (strongly) monotonic or co-coercive w.r.t. $y$ (Gorbunov et al., 2022; Cai et al., 2022; Mahdavinia et al., 2022). In the context of SAM optimization, $x$ corresponds to $\boldsymbol{w}$ and $y$ corresponds to $\boldsymbol{\epsilon}$. Obviously, these assumptions do not hold for deep networks. On the other hand, the following analysis does not need to assume convex loss, and only uses the common assumptions in smooth and non-convex analysis for stochastic gradient methods. Specifically, Assumptions 4.3 and 4.4 below follow from (Andriushchenko & Flammarion, 2022; Bottou et al., 2018; Cutkosky & Orabona, 2019), while Assumption 4.5 follows (Bottou et al., 2018; Hazan & Kale, 2014; Huang et al., 2021), which is used in the proof of convergence analysis for SAM (Mi et al., 2022; Dai et al., 2023; Zhang et al., 2023; Yue et al., 2023).

**Assumption 4.3.** *(Bounded variance) There exists $\sigma \geq 0$ s.t. $\mathbb{E}_{i\sim\mathcal{U}([1,n])}\left[\|\nabla\ell_i(\boldsymbol{w}) - \nabla L(\boldsymbol{w})\|^2\right] \leq \sigma^2$ for all $i \sim \mathcal{U}([1,n])$ (the uniform distribution over $\{1,2,\dots,n\}$, where $n$ is the total number of samples).*

**Assumption 4.4.** *($\beta$-smoothness) There exists $\beta \geq 0$ s.t. $\|\nabla\ell_i(\boldsymbol{w}) - \nabla\ell_i(\boldsymbol{v})\| \leq \beta\|\boldsymbol{w} - \boldsymbol{v}\|$ for all $\boldsymbol{w}, \boldsymbol{v} \in \mathbb{R}^m$ and $i = 1,2,\dots,n$.*

**Assumption 4.5.** *(Uniformly Bounded Gradient) There exists $G \geq 0$ s.t. $\mathbb{E}_{i\sim\mathcal{U}([1,n])}\|\ell_i(\boldsymbol{w})\|^2 \leq G^2$.*

The following provide convergence rates on EG-SAM, OG-SAM, and AO-SAM respectively.

**Theorem 4.6.** *Assume that $\eta_t = \min\left(\frac{1}{2\beta}, \frac{1}{\sqrt{T}}\right)$, $\rho_t = \frac{1}{\sqrt{T}}$ in Algorithm 1, then EG-SAM satisfies $\frac{1}{T}\sum_{t=0}^T \mathbb{E}\|\nabla_{\boldsymbol{w}_t} L(\boldsymbol{w}_t)\|^2 = O\left(\frac{1}{\sqrt{T}} + \frac{1}{\sqrt{T}b}\right)$.*

**Theorem 4.7.** *Assume that $\rho_t = \min\left(\frac{1}{\sqrt{T}}, \frac{1}{\beta}\right)$, $\eta_t = \min\left(\frac{1}{\sqrt{T}}, \frac{1}{\beta}\right)$ in Algorithm 2, then OG-SAM satisfies: $\frac{1}{T}\sum_{t=0}^T \mathbb{E}\|\nabla_{\boldsymbol{w}_t} L(\boldsymbol{w}_t)\|^2 = O\left(\frac{1}{\sqrt{T}} + \frac{1}{\sqrt{T}b}\right)$.*

**Theorem 4.8.** *Assume that $\rho_t = \min\left(\frac{1}{\sqrt{T}}, \frac{1}{\beta}\right)$, $\eta_t = \min\left(\frac{1}{\sqrt{T}}, \frac{1}{2\beta}\right)$ in Algorithm 3, then AO-SAM satisfies: $\frac{1}{T}\sum_{t=0}^T E\|\nabla_{\boldsymbol{w}_t} L(\boldsymbol{w}_t)\|^2 = O\left(\frac{1}{\sqrt{T}} + \frac{1}{\sqrt{T}b}\right)$.*

In summary, EG-SAM, OG-SAM, AO-SAM have the same $O(\frac{1}{\sqrt{T}} + \frac{1}{\sqrt{T}b})$ rate as SAM (Andriushchenko & Flammarion, 2022) and its variant AESAM (Jiang et al., 2023), and is faster than the $O(\log T/\sqrt{T})$ rate of GSAM (Zhuang et al., 2022) and SSAM (Mi et al., 2022).

---

[2]Non-degenerate saddle point means the Hessian matrix at that point is invertible.

# 5 EXPERIMENTS

In this section, we empirically demonstrate the performance of the proposed methods on a number of benchmark datasets. Recall that the training speed is mainly determined by how often the SAM update is used. As in (Jiang et al., 2023), we evaluate efficiency by measuring the fraction of SAM updates used: $\%\text{SAM} \equiv 100 \times (\sum_{t=1}^{T} \#\{\text{SAMs}\} \text{ used at epoch } t)/T$, where $T$ is the total number of epochs and is the same for all methods.

Table 1: Testing accuracy and fraction of SAM updates (%SAM) on *CIFAR-10* using *ResNet-18*. The best accuracy is in bold.

|  | accuracy | %SAM |
|---|---|---|
| SAM | 96.52 ±0.12 | 100.0 ±0.0 |
| SAM+EG | 96.45 ±0.05 | 200.0 ±0.0 |
| SAM+OG | 96.45 ±0.03 | 100.0 ±0.0 |
| SAM+EG-AMO | 92.57 ±0.08 | 200.0 ±0.0 |
| EG-SAM | **96.86** ±0.01 | 150.0 ±0.0 |
| OG-SAM | 96.79 ±0.02 | 100.0 ±0.0 |
| AO-SAM | 96.82 ±0.04 | 61.1 ±0.0 |

Table 2: Testing accuracy and fraction of SAM updates (%SAM) on *CIFAR-100* using *ResNet-18*. The best accuracy is in bold.

|  | accuracy | %SAM |
|---|---|---|
| SAM | 80.17 ±0.05 | 100.0 ±0.0 |
| SAM+EG | 79.91 ±0.16 | 200.0 ±0.0 |
| SAM+OG | 79.92 ±0.08 | 100.0 ±0.0 |
| SAM+EG-AMO | 74.65 ±0.07 | 200.0 ±0.0 |
| EG-SAM | **80.89** ±0.12 | 150.0 ±0.0 |
| OG-SAM | 80.76 ±0.15 | 100.0 ±0.0 |
| AO-SAM | 80.70 ±0.14 | 61.2 ±0.0 |

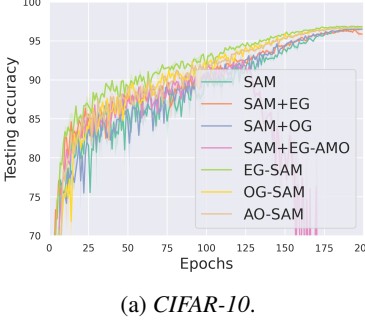

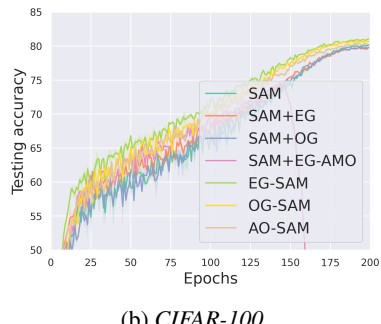

(a) *CIFAR-10*.

(b) *CIFAR-100*.

Figure 2: Convergence on *CIFAR-10* and *CIFAR-100* (with *ResNet-18* backbone).

## 5.1 *CIFAR-10* AND *CIFAR-100*

**Setup.** In this section, we perform experiments on the popular image classification datasets *CIFAR-10* and *CIFAR-100* (Krizhevsky et al., 2009). $10\%$ of the training set is used for validation. Following the SAM literature (Jiang et al., 2023; Mi et al., 2022; Kwon et al., 2021), we use the commonly-used *ResNet-18* (He et al., 2016), *ResNet-32* (He et al., 2016), and *WideResNet-28-10* (Zagoruyko & Komodakis, 2016).

Following the setup in (Jiang et al., 2023; Foret et al., 2021), we use batch size 128, initial learning rate 0.1, cosine learning rate schedule (Loshchilov & Hutter, 2017), and SGD optimizer. The number of training epochs is 200 for all experiments. For the proposed methods, we select $\rho \in \{0.01, 0.05, 0.08, 0.1, 0.5, 0.8, 1, 1.5, 1.8, 2\}$ by using *CIFAR-10*'s validation set on *ResNet-18*. The selected $\rho$ is then directly used on *CIFAR-100* and the other backbones. For the $c_t$ schedule in (5), since different SAM variants yield different %SAM's, we vary the hyper-parameters $(\kappa_1, \kappa_2)$ so that the %SAM obtained by AO-SAM matches their %SAM values.

**Comparison among EG-SAM, OG-SAM, AO-SAM and SAM**. First, we compare the proposed EG-SAM (Extra Gradient SAM), OG-SAM (Optimistic Gradient SAM), AO-SAM (Adaptive Optimistic SAM) with SAM and its three variants: (i) SAM+EG, which directly applies the EG updates (7) on SAM's objective (6); (ii) SAM+OG, which directly applies the OG updates ((7) that replaces the update on $\bar{z}_t$ with (8)) on (6); (3) SAM+EG-AMO (SAM+EG approximated max oracle), which uses (13) as the update scheme . Experiments are performed on *CIFAR-10* and *CIFAR-100* with the *ResNet-18* backbone, and repeated 5 times with different random seeds.

Tables 1 and 2 shows the testing accuracy versus %SAM. Figure 2 shows the testing accuracies (results on the loss and training accuracies are shown in Figure 4 in Appendix B). Note that as SAM+EG takes two SAM steps ((9), (10) and (11), (12)) in every epoch, its %SAM is 200. Similarly,

Table 3: Testing accuracy and fraction of SAM updates on *CIFAR-10* with different levels of label noise. Results of ERM, SAM, and ESAM with *ResNet-18* and *ResNet-32* are from (Jiang et al., 2023). † means (Jiang et al., 2023) do not provide standard derivation for that baseline. Other baseline results are obtained with the authors' provided code. The best accuracy is in bold.

| | | noise $= 20\%$ | | noise $= 40\%$ | | noise $= 60\%$ | | noise $= 80\%$ | |
| | | accuracy | %SAM | accuracy | %SAM | accuracy | %SAM | accuracy | %SAM |
|---|---|---|---|---|---|---|---|---|---|
| | ERM | $87.92^{\dagger}$ | 0.0 | $70.82^{\dagger}$ | 0.0 | $49.61^{\dagger}$ | 0.0 | $28.23^{\dagger}$ | 0.0 |
| *ResNet-18* | SAM (Foret et al., 2021) | $94.80^{\dagger}$ | 100.0 | $91.50^{\dagger}$ | 100.0 | $88.15^{\dagger}$ | 100.0 | $77.40^{\dagger}$ | 100.0 |
| | ESAM (Du et al., 2022a) | $94.19^{\dagger}$ | 100.0 | $91.46^{\dagger}$ | 100.0 | $81.30^{\dagger}$ | 100.0 | $15.00^{\dagger}$ | 100.0 |
| | ASAM (Kwon et al., 2021) | $91.17 \pm 0.19$ | 100.0 | $87.38 \pm 0.61$ | 100.0 | $83.22 \pm 0.41$ | 100.0 | $71.03 \pm 0.88$ | 100.0 |
| | OG-SAM | $\mathbf{95.12} \pm 0.12$ | 100.0 | $92.16 \pm 0.35$ | 100.0 | $88.45 \pm 0.53$ | 100.0 | $77.47 \pm 0.65$ | 100.0 |
| | SS-SAM (Zhao, 2022) | $94.61 \pm 0.16$ | 60.0 | $91.81 \pm 0.13$ | 60.0 | $78.67 \pm 0.42$ | 60.0 | $62.94 \pm 1.01$ | 60.0 |
| | AE-SAM (Jiang et al., 2023) | $92.13 \pm 0.14$ | 61.4 | $86.02 \pm 0.62$ | 61.4 | $75.95 \pm 1.30$ | 61.4 | $67.28 \pm 1.66$ | 61.4 |
| | AO-SAM | $95.02 \pm 0.04$ | 61.2 | $\mathbf{92.62} \pm 0.18$ | 61.3 | $\mathbf{89.36} \pm 0.12$ | 61.2 | $\mathbf{78.12} \pm 0.38$ | 61.2 |
| | ERM | $87.43^{\dagger}$ | 0.0 | $70.82^{\dagger}$ | 0.0 | $46.26^{\dagger}$ | 0.0 | $29.00^{\dagger}$ | 0.0 |
| *ResNet-32* | SAM (Foret et al., 2021) | $95.08^{\dagger}$ | 100.0 | $91.01^{\dagger}$ | 100.0 | $88.90^{\dagger}$ | 100.0 | $77.32^{\dagger}$ | 100.0 |
| | ESAM (Du et al., 2022a) | $93.42^{\dagger}$ | 100.0 | $91.63^{\dagger}$ | 100.0 | $82.73^{\dagger}$ | 100.0 | $10.09^{\dagger}$ | 100.0 |
| | ASAM (Kwon et al., 2021) | $92.04 \pm 0.09$ | 100.0 | $88.83 \pm 0.11$ | 100.0 | $83.90 \pm 0.56$ | 100.0 | $75.64 \pm 0.75$ | 100.0 |
| | OG-SAM | $95.25 \pm 0.04$ | 100.0 | $\mathbf{92.11} \pm 0.07$ | 100.0 | $88.36 \pm 0.22$ | 100.0 | $77.61 \pm 0.39$ | 100.0 |
| | SS-SAM (Zhao, 2022) | $95.03 \pm 0.23$ | 60.0 | $90.59 \pm 0.30$ | 60.0 | $87.22 \pm 0.46$ | 60.0 | $48.89 \pm 1.02$ | 60.0 |
| | AE-SAM (Jiang et al., 2023) | $92.04 \pm 0.27$ | 61.3 | $86.83 \pm 0.49$ | 61.3 | $73.90 \pm 0.44$ | 61.2 | $67.64 \pm 1.34$ | 61.3 |
| | AO-SAM | $\mathbf{95.32} \pm 0.12$ | 61.2 | $91.73 \pm 0.65$ | 61.2 | $\mathbf{89.40} \pm 0.44$ | 61.2 | $\mathbf{77.78} \pm 0.84$ | 61.2 |
| | ERM | $90.07 \pm 0.36$ | 0.0 | $86.02 \pm 0.33$ | 0.0 | $80.98 \pm 0.52$ | 0.0 | $67.67 \pm 0.72$ | 0.0 |
| *WideResNet-28-10* | SAM (Foret et al., 2021) | $94.47 \pm 0.12$ | 100.0 | $91.74 \pm 0.04$ | 100.0 | $88.35 \pm 0.21$ | 100.0 | $71.37 \pm 1.55$ | 100.0 |
| | ESAM (Du et al., 2022a) | $95.09 \pm 0.04$ | 100.0 | $89.16 \pm 0.21$ | 100.0 | $42.64 \pm 0.55$ | 100.0 | $20.14 \pm 0.69$ | 100.0 |
| | ASAM (Kwon et al., 2021) | $91.25 \pm 0.16$ | 100.0 | $88.08 \pm 0.07$ | 100.0 | $83.45 \pm 0.12$ | 100.0 | $71.44 \pm 0.46$ | 100.0 |
| | OG-SAM | $95.31 \pm 0.06$ | 100.0 | $92.67 \pm 0.13$ | 100.0 | $88.37 \pm 0.58$ | 100.0 | $\mathbf{77.86} \pm 1.83$ | 100.0 |
| | SS-SAM (Zhao, 2022) | $94.47 \pm 0.09$ | 60.0 | $91.90 \pm 0.11$ | 60.0 | $88.43 \pm 0.37$ | 60.0 | $74.64 \pm 0.79$ | 60.0 |
| | AE-SAM (Jiang et al., 2023) | $93.49 \pm 0.14$ | 61.3 | $90.36 \pm 0.12$ | 61.3 | $85.95 \pm 0.47$ | 61.3 | $71.21 \pm 1.56$ | 61.3 |
| | AO-SAM | $\mathbf{95.52} \pm 0.24$ | 61.1 | $\mathbf{92.68} \pm 0.10$ | 61.2 | $89.29 \pm 0.28$ | 61.2 | $77.13 \pm 0.72$ | 61.2 |

for SAM+EG-AMO, its %SAM is 200; and for EG-SAM, its %SAM is 150. As can be seen, though EG-SAM has the highest accuracy, it is also the third slowest. On the other hand, OG-SAM is as fast as SAM, but is more accurate. Similarly, AO-SAM is as accurate as OG-SAM, but is faster. Hence, we will only focus on AO-SAM in the sequel. Moreover, comparing with losses and accuracies, there exists monotonic relationship between training (resp. testing) accuracy and training (resp. testing) loss for EG-SAM, OG-SAM, and AO-SAM.

Note that SAM+OG only outperforms SAM on *CIFAR-10* slightly, and SAM+EG performs even worse than SAM. On *CIFAR-100*, SAM+EG and SAM+OG perform worse than SAM. This is because EG and OG belong to the GDA family, and a direct use on SAM inherits their slow convergence (Section 3.4). This can also be seen from the convergence plots in Figure 2, which shows that SAM+OG and SAM+EG have slower convergence. SAM+EG-AMO also performs worse than SAM on *CIFAR-10* and *CIFAR-100*, indicating that directly using minimization based extra-gradient does not perform well due to the approximation. On the contrary, EG-SAM, OG-SAM, and AO-SAM have relatively fast convergence rate on both datasets.

**Comparison with Baselines in *CIFAR-10* and *CIFAR-100*.**

Following (Jiang et al., 2023), we compare AO-SAM and OG-SAM with: (i) ERM; (ii) SAM (Foret et al., 2021); and its variants (iii) ESAM (Du et al., 2022a), (iv) ASAM (Kwon et al., 2021), (v) SS-SAM (Zhao, 2022), and (vi) AE-SAM (Jiang et al., 2023). As different SAM variants yield different %SAM's, we vary the $(\kappa_1, \kappa_2)$ values in (5) for AE-SAM and AO-SAM so as to attain comparable %SAM values for fairer comparison. We also study the robustness of the various methods to label noise. Specifically, a certain fraction (20%, 40%, 60% and 80%) of the training labels in *CIFAR-10* and *CIFAR-100* are randomly flipped. Experiments are repeated 5 times with different random seeds.

Table 3 shows the results on *CIFAR-10* corrupted with label noise. Results on *CIFAR-100* are in table 6 in Appendix B. As can be seen, AO-SAM and OG-SAM outperform all baselines at all label noise ratios on both data sets. Moreover, the accuracy improvement gets larger as the label noise ratio increases. This demonstrates the superiority of AO-SAM and OG-SAM particularly in difficult learning environments that requires better generalization ability.

Results on the datasets without label noise are shown in Table 7 in Appendix B. As can be seen, OG-SAM and AO-SAM are still consistently more accurate than the baselines on both datasets and all models.

## 5.2 *ImageNet*

In this experiment, we perform experiment on the *ImageNet* dataset using *ResNet-50* (He et al., 2016). The batch size is 512, initial learning rate is 0.1, cosine learning rate schedule, and the number of training epochs is 90. the other experimental setup are the same as in Section 5.1. The experiment is repeated 3 times with different random seeds. OG-SAM is not compared here, as previous experiments show that it does not perform better than AO-SAM but takes more %SAM. Table 4 shows the testing accuracy and %SAM. As can be seen, the proposed AO-SAM again outperforms all the baselines.

Table 4: Testing accuracy (mean and standard deviation) and fraction of SAM updates (%SAM) on *ImageNet* using *ResNet-50*. Results of SAM and ESAM are from (Jiang et al., 2023), ASAM is from (Kwon et al., 2021), and other baselines are obtained by the authors' provide code.

Table 5: Eigenvalues of the Hessian on *CIFAR-10* with *ResNet18*.

|  | accuracy | %SAM |
|---|---|---|
| ERM | 77.11 $\pm_{0.14}$ | 0.0 $\pm_{0.0}$ |
| SAM (Foret et al., 2021) | 77.47 $\pm_{0.12}$ | 100.0 $\pm_{0.0}$ |
| ESAM (Du et al., 2022a) | 77.25 $\pm_{0.75}$ | 100.0 $\pm_{0.0}$ |
| ASAM (Kwon et al., 2021) | 76.63 $\pm_{0.18}$ | 100.0 $\pm_{0.0}$ |
| AO-SAM | **77.68** $\pm_{0.04}$ | 61.1 $\pm_{0.0}$ |
| SS-SAM (Zhao, 2022) | 77.41 $\pm_{0.05}$ | 60.0 $\pm_{0.0}$ |
| AE-SAM (Jiang et al., 2023) | 77.46 $\pm_{0.07}$ | 61.3 $\pm_{0.0}$ |

|  | $\lambda_1$ | $\lambda_1/\lambda_5$ |
|---|---|---|
| ERM | 88.8 | 3.3 |
| SAM | 29.6 | 3.3 |
| EG-SAM | 10.8 | 1.8 |
| OG-SAM | 13.1 | 2.0 |
| AO-SAM | 11.1 | 1.8 |

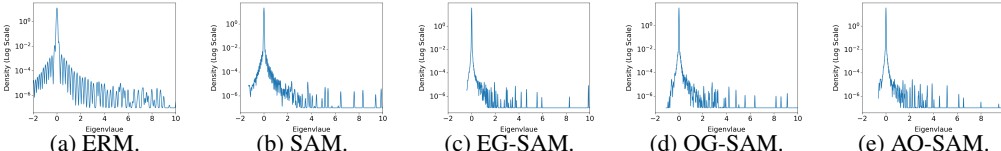

(a) ERM.  (b) SAM.  (c) EG-SAM.  (d) OG-SAM.  (e) AO-SAM.

Figure 3: Hessian spectra obtained by ERM, SAM, EG-SAM, OG-SAM, and AO-SAM on *CIFAR-10* with *ResNet18*.

## 5.3 FLAT MINIMA

In this section, we compare the abilities of ERM, SAM, OG-SAM and AO-SAM to converge to flat minima. Following (Mi et al., 2022; Foret et al., 2021), we illustrate this by examining the eigenvalue spectrum of the Hessian at the converged solution. Experiments are performed on *CIFAR-10* with the *ResNet-18* backbone. As can be seen from Figure 3, the Hessian's eigenvalues of EG-SAM, OG-SAM and AO-SAM are smaller than those of ERM and SAM, indicating that the loss landscapes at both OG-SAM's and AO-SAM's converged solutions are flatter compared to SAM and ERM. As in (Foret et al., 2021; Mi et al., 2022), Table 5 shows the largest eigenvalue of the Hessian ($\lambda_1$) and the ratio $\lambda_1/\lambda_5$ (where $\lambda_5$ is the 5th largest eigenvalue). As can be seen, EG-SAM, OG-SAM and AO-SAM have smaller $\lambda_1$ and $\lambda_1/\lambda_5$ than ERM and SAM, again indicating that they have flatter minima than SAM and ERM. This agrees with Section 4.1 that EG-SAM converges to flat minima, and Table 7 where EG-SAM, AO-SAM, and OG-SAM have higher accuracies than SAM and ERM.

## 6 CONCLUSION

In this paper, we integrate the lookahead mechanism, which has been proven effective in game theory and optimization, into SAM. Lookahead enables the model to gain more information about the loss landscape, thus alleviating the problem of convergence instability in SAM's minimax optimization process. Theoretical results show that the proposed method can converge to a stationary point and is not easy to be trapped in saddle points. Experiments on standard benchmark datasets also verify that the proposed method outperforms the SOTAs, and converges more effectively to flat minima.

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

# A PROOFS

## A.1 PROOFS FOR EG-SAM AND SAM ODE

Before we dive into our proof, we briefly discuss the connection between the ODE of SAM (19) and EG-SAM (20), and the gradient descent scheme for SAM (3) and EG-SAM (14-16):

Firstly, rewrite the (3) as:

$$
\frac{\boldsymbol{w}_t - \boldsymbol{w}_{t-1}}{\eta} = -\nabla_{\boldsymbol{w}_{t-1}} L \left( \boldsymbol{w}_{t-1} + \boldsymbol{\epsilon}_{t-1} \Big|_{\boldsymbol{\epsilon}_{t-1} = \frac{\rho \nabla_{\boldsymbol{w}_{t-1}} L(\boldsymbol{w}_{t-1})}{\|\nabla_{\boldsymbol{w}_{t-1}} L(\boldsymbol{w}_{t-1})\|}} \right),
$$

Then, replacing $\frac{\boldsymbol{w}_t - \boldsymbol{w}_{t-1}}{\eta}$ by $\frac{d\boldsymbol{w}_t}{dt}$ [3]. Note that $\nabla_{\boldsymbol{w}} L(\boldsymbol{w}) = H\boldsymbol{w}$ when $L = \boldsymbol{w}^\top H\boldsymbol{w}$. Putting them together, we obtain our result.

EG-SAM can be obtained by using similar approach to (14-16). Now we start our proof.

**Lemma A.1.** *(SAM ODE, Lemma C.6 (Compagnoni et al., 2023)) In terms of (18), for all $\rho > 0$, if $H$ is PSD, the origin is (locally) asymptotically stable. Additionally, if $H$ is not PSD, and $\|H\boldsymbol{w}_\tau\| \leq -\rho\lambda_i, \forall i$, then the origin is still (locally) asymptotically stable.*

**Lemma A.2.** *(EG-SAM ODE) In terms of (18), if $\rho \geq \|H\boldsymbol{w}_\tau\| \left( \eta_\tau - \frac{1}{\lambda_i} \right), \forall i, \tau$, the origin is (locally) asymptotically stable.*

*Proof.* Let $V(x) := \frac{\boldsymbol{w}_\tau^\top K \boldsymbol{w}_\tau}{2}$ be the Lyapunov function of EG-SAM, where $K$ is a diagonal matrix with positive eigenvalues $(k_1, \cdots, k_d)$. Therefore, we have

$$
V(\boldsymbol{w}_\tau) = \frac{1}{2} \sum_{i=1}^d k_i \left( \boldsymbol{w}_\tau^i \right)^2 > 0,
$$

And

$$
\dot{V}(\boldsymbol{w}_\tau) = \sum_{i=1}^d k_i \boldsymbol{w}_\tau^i \frac{d\boldsymbol{w}_\tau^i}{d\tau} \stackrel{(a)}{=} \sum_{i=1}^d q_{ii}^2 k_i \boldsymbol{w}_\tau^i \left( -\lambda_i \left( \boldsymbol{w}_\tau^i + \frac{\rho H \boldsymbol{w}_\tau^i}{\|H\boldsymbol{w}_\tau\|} - \eta_\tau \lambda_i \boldsymbol{w}_\tau^i \right) \right)
$$

$$
= \sum_{i=1}^d q_{ii}^2 k_i (-\lambda_i) \left( 1 + \frac{\rho \lambda_i}{\|H\boldsymbol{w}_\tau\|} - \eta_\tau \lambda_i \right) \left( \boldsymbol{w}_\tau^i \right)^2 d\tau
$$

(a) is because the $H$ is symmetry and can be decomposed as $H = Q^\top \Lambda Q$ (Theorem 5.11 in Apostol (1991)), where $Q$ is an orthogonal matrix with $q_{ij}$ as its $i$ row $j$ column element, and $\Lambda$ is diagonal matrix of the eigenvalues of $H$. Then, $\frac{d\boldsymbol{w}_\tau^i}{d\tau}$ can be written as $\frac{d\boldsymbol{w}_\tau^i}{d\tau} = \left( -\lambda_i \left( \boldsymbol{w}_\tau^i + \frac{\rho \lambda_i \boldsymbol{w}_\tau^i}{\|H\boldsymbol{w}_\tau\|} - \eta_\tau \lambda_i \boldsymbol{w}_\tau^i \right) \right) q_{ii}^2$.

For the term $q_{ii}^2 k_i \lambda_i \left( 1 + \frac{\rho \lambda_i}{\|H\boldsymbol{w}_\tau\|} - \eta_\tau \lambda_i \boldsymbol{w}_\tau^i \right) \left( \boldsymbol{w}_\tau^i \right)^2$, when $\lambda_i > 0$, to ensure $-q_{ii}^2 k_i \lambda_i \left( 1 + \frac{\rho \lambda_i}{\|H\boldsymbol{w}_\tau\|} - \eta_\tau \lambda_i \boldsymbol{w}_\tau^i \right) \left( \boldsymbol{w}_\tau^i \right)^2 \leq 0$, we should have $1 + \frac{\rho \lambda_i}{\|H\boldsymbol{w}_\tau\|} - \eta_\tau \lambda_i \geq 0$, and thus $\rho \geq \|H\boldsymbol{w}_\tau\| \left( \eta_\tau - \frac{1}{\lambda_i} \right)$.

Similarly, when $\lambda_i < 0$, to ensure $-q_{ii}^2 k_i \lambda_i \left( 1 + \frac{\rho \lambda_i}{\|H\boldsymbol{w}_\tau\|} - \eta_\tau \lambda_i \boldsymbol{w}_\tau^i \right) \left( \boldsymbol{w}_\tau^i \right)^2 \leq 0$, we should have $1 + \frac{\rho \lambda_i}{\|H\boldsymbol{w}_\tau\|} - \eta_\tau \lambda_i \leq 0$, and thus $\rho \geq \|H\boldsymbol{w}_\tau\| \left( \eta_\tau - \frac{1}{\lambda_i} \right)$.

---

[3] The justification of this approximation can be found in https://francisbach.com/gradient-flows/

Therefore, when $\rho \geq \|H\boldsymbol{w}_\tau\| \left(\eta_\tau - \frac{1}{\lambda_i}\right), \forall i$, we always have $V(\boldsymbol{w}_\tau) > 0$ and $\dot{V}(\boldsymbol{w}_\tau) \leq 0$. According to Theorem 1.1 (Mao, 2007), the dynamics of $\boldsymbol{w}_\tau$ is bounded inside this set and cannot diverge if $V(\boldsymbol{w}_\tau) > 0$ and $\dot{V}(\boldsymbol{w}_\tau) \leq 0$.

$\square$

**Proposition A.3.** *For non-degenerated saddle point, EG-SAM has smaller ROA than SAM.*

*Proof.* Recall that non-degenerated saddle point has both positive and negative eigenvalues (Theorem 9.6. in (Apostol, 1991)). According to Lemma A.1, the ROA for SAM is: $\left\{\boldsymbol{w}_t | \rho \geq -\frac{\|H\boldsymbol{w}_t\|}{\lambda_{\min}}\right\}$. The ROA for EG-SAM is: $\left\{\boldsymbol{w}_t | \rho \geq \|H\boldsymbol{w}_t\| \left(\eta_\tau - \frac{1}{\lambda_{\min}}\right)\right\}$, according to Lemma A.2. Since $\left(\eta - \frac{1}{\lambda_{\min}}\right) > \left(-\frac{1}{\lambda_{\min}}\right)$, we have: $\left\{\boldsymbol{w}_t | \rho \geq \|H\boldsymbol{w}_t\| \left(\eta_\tau - \frac{1}{\lambda_{\min}}\right)\right\} \subset \left\{\boldsymbol{w}_t | \rho \geq -\frac{\|H\boldsymbol{w}_t\|}{\lambda_{\min}}\right\}$, which implies EG-SAM has smaller ROA than SAM.

$\square$

**Proposition A.4.** *For non-degenerated stationary points $\boldsymbol{w}_1^*$ and $\boldsymbol{w}_2^*$ with largest eigenvalues $\lambda_{\max}^1, \lambda_{\max}^2$, if $\lambda_{\max}^1 > \lambda_{\max}^2$, then $\boldsymbol{w}_1^*$ has smaller ROA than $\boldsymbol{w}_2^*$.*

*Proof.* Recall that non-degenerated stationary point has all positive eigenvalues (Theorem 9.6. in (Apostol, 1991)). The ROA for EG-SAM is: $\left\{\boldsymbol{w}_t | \rho \geq \|H\boldsymbol{w}_t\| \left(\eta_\tau - \frac{1}{\lambda_{\max}}\right)\right\}$, according to Lemma A.2. Since $\lambda_{\max}^1 > \lambda_{\max}^2$, we have: $\left\{\boldsymbol{w}_t | \rho \geq \|H\boldsymbol{w}_t\| \left(\eta_\tau - \frac{1}{\lambda_{\max}^1}\right)\right\} \subset \left\{\boldsymbol{w}_t | \rho \geq \|H\boldsymbol{w}_t\| \left(\eta_\tau - \frac{1}{\lambda_{\max}^2}\right)\right\}$, which implies that $\boldsymbol{w}_1^*$ has smaller ROA than $\boldsymbol{w}_2^*$.

$\square$

## A.2 EG-SAM CONVERGENCE

Let $L_t(\boldsymbol{w}) := \frac{1}{b} \sum_{i \in I_t} \ell_i(\boldsymbol{w})$ be the mini-batch version of $L(\boldsymbol{w})$ at epoch $t$, where $\boldsymbol{w} \in \mathbb{R}^m$, $I_t$ is the mini-batch, and $\ell_i(\boldsymbol{w})$ is the loss for sample $i$. Let $\boldsymbol{w}_{t-1/2} := \hat{\boldsymbol{w}}_t + \hat{\boldsymbol{\epsilon}}_t = \boldsymbol{w}_{t-1} + \rho \frac{\nabla L_t(\boldsymbol{w}_{t-1})}{\|\nabla L_t(\boldsymbol{w}_{t-1})\|} - \eta \nabla L_t(\boldsymbol{w}_{t-1})$. Note that the update scheme of EG-SAM can be rewritten as: $\boldsymbol{w}_t = \boldsymbol{w}_{t-1} - \eta_t \nabla L_t(\boldsymbol{w}_{t-1/2})$. In the following, we assume that $\rho, \eta, \beta > 0$.

**Lemma A.5.**
$$\left\langle \nabla L \left(\boldsymbol{w} + \rho \frac{\nabla L(\boldsymbol{w})}{\|\nabla L(\boldsymbol{w})\|} - \eta \nabla L(\boldsymbol{w})\right), \nabla L(\boldsymbol{w}) \right\rangle \geq (1 + \beta\eta)\|\nabla L(\boldsymbol{w})\|^2 - \beta\rho\|\nabla L(\boldsymbol{w})\|.$$

*Proof.*

$$\left\langle \nabla L(\boldsymbol{w} + \rho \frac{\nabla L(\boldsymbol{w})}{\|\nabla L(\boldsymbol{w})\|} - \eta \nabla L(\boldsymbol{w})), \nabla L(\boldsymbol{w}) \right\rangle$$

$$= \left\langle \nabla L(\boldsymbol{w} + (\rho - \eta\|\nabla L(\boldsymbol{w})\|) \frac{\nabla L(\boldsymbol{w})}{\|\nabla L(\boldsymbol{w})\|}) - \nabla L(\boldsymbol{w}), \nabla L(\boldsymbol{w}) \right\rangle + \|\nabla L(\boldsymbol{w})\|^2$$

$$= \frac{\|\nabla L(\boldsymbol{w})\|}{\rho - \eta\|\nabla L(\boldsymbol{w})\|} \left\langle \nabla L(\boldsymbol{w} + (\rho - \eta\|\nabla L(\boldsymbol{w})\|) \frac{\nabla L(\boldsymbol{w})}{\|\nabla L(\boldsymbol{w})\|}) - \nabla L(\boldsymbol{w}), \frac{(\rho - \eta\|\nabla L(\boldsymbol{w})\|)}{\|\nabla L(\boldsymbol{w})\|} \nabla L(\boldsymbol{w}) \right\rangle$$
$$+ \|\nabla L(\boldsymbol{w})\|^2$$

$$\overset{(a)}{\geq} (1 - \frac{\beta(\rho - \eta\|\nabla L(\boldsymbol{w})\|)}{\|\nabla L(\boldsymbol{w})\|})\|\nabla L(\boldsymbol{w})\|^2$$

$$\geq (1 + \eta\beta)\|\nabla L(\boldsymbol{w})\|^2 - \beta\rho\|\nabla L(\boldsymbol{w})\|.$$

The first equation is based on the fact that $\langle \nabla L(w), \nabla L(w) \rangle = \|\nabla L(w)\|^2$, while (a) uses the co-coercivity property of the smooth function $L$.

$\square$

**Lemma A.6.**

$$\mathbb{E}\langle\nabla L_t(\boldsymbol{w}+\rho\nabla L_t(\boldsymbol{w})/\|\nabla L_t(\boldsymbol{w})\|-\eta\nabla L_t(\boldsymbol{w})),\nabla L(\boldsymbol{w})\rangle \geq (\frac{1}{2}+\eta\beta)\|\nabla L(\boldsymbol{w})\|^2-\beta\rho\|\nabla L(\boldsymbol{w})\|-\frac{\beta^2\sigma^2\eta^2}{2b}-\rho^2\beta^2.$$

*Proof.* Note that

$$\left\langle\nabla L_t\left(\boldsymbol{w}+\rho\frac{\nabla L_t(\boldsymbol{w})}{\|\nabla L_t(\boldsymbol{w})\|}-\eta\nabla L_t(\boldsymbol{w})\right),\nabla L(\boldsymbol{w})\right\rangle$$
$$=\left\langle\nabla L_t\left(\boldsymbol{w}+\rho\frac{\nabla L_t(\boldsymbol{w})}{\|\nabla L_t(\boldsymbol{w})\|}-\eta\nabla L_t(\boldsymbol{w})\right)-\nabla L_t(\boldsymbol{w}+\rho\frac{\nabla L(\boldsymbol{w})}{\|\nabla L(\boldsymbol{w})\|}-\eta\nabla L(\boldsymbol{w})),\nabla L(\boldsymbol{w})\right\rangle$$
$$-\left\langle-\nabla L_t(\boldsymbol{w}+\rho\frac{\nabla L(\boldsymbol{w})}{\|\nabla L(\boldsymbol{w})\|}-\eta\nabla L(\boldsymbol{w})),\nabla L(\boldsymbol{w})\right\rangle.$$

In the following, we bound the first term and second term of the RHS separately. For the first term,

$$-\mathbb{E}\left\langle\nabla L_t\left(\boldsymbol{w}+\rho\frac{\nabla L_t(\boldsymbol{w})}{\|\nabla L_t(\boldsymbol{w})\|}-\eta\nabla L_t(\boldsymbol{w})\right)-\nabla L_t(\boldsymbol{w}+\rho\frac{\nabla L(\boldsymbol{w})}{\|\nabla L(w)\|}-\eta\nabla L(\boldsymbol{w})),\nabla L(w)\right\rangle$$

$$\overset{(a)}{\leq} \frac{1}{2}\mathbb{E}\|\nabla L_t\left(\boldsymbol{w}+\rho\frac{\nabla L_t(\boldsymbol{w})}{\|\nabla L_t(\boldsymbol{w})\|}-\eta\nabla L_t(\boldsymbol{w})\right)-\nabla L_t(\boldsymbol{w}+\rho\frac{\nabla L(\boldsymbol{w})}{\|\nabla L(\boldsymbol{w})\|}-\eta\nabla L(\boldsymbol{w}))\|^2$$
$$+\frac{1}{2}\|\nabla L(\boldsymbol{w})\|^2$$

$$\leq \frac{\beta^2}{2}\mathbb{E}\|\rho\frac{\nabla L_t(\boldsymbol{w})}{\|\nabla L_t(\boldsymbol{w})\|}-\eta\nabla L_t(\boldsymbol{w})-(\rho\frac{\nabla L(\boldsymbol{w})}{\|\nabla L(\boldsymbol{w})\|}-\eta\nabla L(\boldsymbol{w}))\|^2+\frac{1}{2}\|\nabla L(\boldsymbol{w})\|^2$$

$$\overset{(b)}{\leq} \frac{\beta^2\sigma^2\eta^2}{2b}+\rho^2\beta^2+\frac{1}{2}\|\nabla L(\boldsymbol{w})\|^2.$$

(a) is based on the Young's inequality, (b) is using the triangle inequality and Assumption 4.4.

For the second term, on using Lemma A.5,

$$\mathbb{E}\left\langle\nabla L_t(\boldsymbol{w}+\rho\frac{\nabla L(\boldsymbol{w})}{\|\nabla L(\boldsymbol{w})\|}-\eta\nabla L(\boldsymbol{w})),\nabla L(\boldsymbol{w})\right\rangle$$
$$= \mathbb{E}\left\langle\nabla L(\boldsymbol{w}+\rho\frac{\nabla L(\boldsymbol{w})}{\|\nabla L(\boldsymbol{w})\|}-\eta\nabla L(\boldsymbol{w})),\nabla L(\boldsymbol{w})\right\rangle$$
$$\geq (1+\beta\eta)\|\nabla L(\boldsymbol{w})\|^2-\beta\rho\|\nabla L(\boldsymbol{w})\|.$$

Combining them together, we obtain the result. $\square$

**Lemma A.7.** *With assumptions 4.3, 4.4 and 4.5, and also assume $\eta\leq\frac{1}{2\beta}$, we have:*

$$\eta\left(\eta\beta-\eta^3\beta^3-2\eta\beta\left(\frac{1}{2}+\eta\beta\right)+\left(\frac{1}{2}+\eta\beta\right)\right)\mathbb{E}\|\nabla L(\boldsymbol{w}_t)\|^2$$

$$\leq \mathbb{E}L(\boldsymbol{w}_t)-\mathbb{E}L(\boldsymbol{w}_{t+1})+\eta^2\beta^3\rho^2+\eta(1-2\eta\beta)\beta\rho G+\eta(1-2\eta\beta)\frac{\beta^2\sigma^2}{2b}$$
$$+2\rho^2\beta^2\eta(1-2\eta\beta)+\eta^2\beta\frac{\sigma^2}{b}.$$

*Proof.* Using the property of smooth function $L$ (Assumption (4.4)), we have:

$$\mathbb{E}L(\boldsymbol{w}_{t+1}) \leq \mathbb{E}L(\boldsymbol{w}_t)-\eta\mathbb{E}\langle\nabla L(\boldsymbol{w}_{t-1/2}),\nabla L(\boldsymbol{w}_t)\rangle+\frac{\eta^2\beta}{2}\mathbb{E}\|\nabla L_t(\boldsymbol{w}_{t-1/2})\|^2$$

$$\leq \mathbb{E}L(\boldsymbol{w}_t)-\eta\mathbb{E}\langle\nabla L(\boldsymbol{w}_{t-1/2}),\nabla L(\boldsymbol{w}_t)\rangle+\frac{\eta^2\beta}{2}\mathbb{E}\|\nabla L_t(\boldsymbol{w}_{t-1/2})-\nabla L(\boldsymbol{w}_{t-1/2})\|^2$$
$$+\frac{\eta^2\beta}{2}\mathbb{E}\|\nabla L(\boldsymbol{w}_{t-1/2})\|^2$$

$$\leq \mathbb{E}L(\boldsymbol{w}_t)-\eta\mathbb{E}\langle\nabla L(\boldsymbol{w}_{t-1/2}),\nabla L(\boldsymbol{w}_t)\rangle+\eta^2\beta\frac{\sigma^2}{2b}+\frac{\eta^2\beta}{2}\mathbb{E}\|\nabla L(\boldsymbol{w}_{t-1/2})\|^2.$$

The first inequality is using the property of smooth function (Assumption (4.4)) and take the expectation on both sides.

Then,

$$
\begin{aligned}
\mathbb{E}L(\boldsymbol{w}_{t+1}) &\overset{(a)}{\leq} \mathbb{E}L(\boldsymbol{w}_t) - \eta^2\beta\mathbb{E}\|\nabla L(\boldsymbol{w}_t)\|^2 + \eta^2\beta\mathbb{E}\|\nabla L(\boldsymbol{w}_{t-1/2}) - \nabla L(\boldsymbol{w}_t)\|^2 \\
&\quad -\eta(1-2\eta\beta)\mathbb{E}\langle\nabla L(\boldsymbol{w}_{t-1/2}), \nabla L(\boldsymbol{w}_t)\rangle + \eta^2\beta\frac{\sigma^2}{2b} \\
&\overset{(b)}{\leq} \mathbb{E}L(\boldsymbol{w}_t) - \eta^2\beta\mathbb{E}\|\nabla L(\boldsymbol{w}_t)\|^2 + \eta^2\beta^3\mathbb{E}\|\boldsymbol{w}_{t-1/2} - \boldsymbol{w}_t\|^2 \\
&\quad -\eta(1-2\eta\beta)\left[-\beta\rho\mathbb{E}\|\nabla L(\boldsymbol{w}_t)\| + \left(\frac{1}{2}+\eta\beta\right)\mathbb{E}\|\nabla L(\boldsymbol{w}_t)\|^2 - \frac{\beta^2\sigma^2\eta^2}{2b} - \rho^2\beta^2\right] \\
&\quad +\eta^2\beta\frac{\sigma^2}{2b} \\
&\leq \mathbb{E}L(\boldsymbol{w}_t) - \eta^2\beta\mathbb{E}\|\nabla L(w_t)\|^2 + \eta^2\beta^3\rho^2 + \eta^4\beta^3\,\mathbb{E}\|\nabla L(\boldsymbol{w}_t)\|^2 \\
&\quad +\eta(1-2\eta\beta)\beta\rho\mathbb{E}\|\nabla L(\boldsymbol{w}_t)\| - \eta(1-2\eta\beta)\left(\frac{1}{2}+\eta\beta\right)\mathbb{E}\|\nabla L(\boldsymbol{w}_t)\|^2 \\
&\quad +\eta^3(1-2\eta\beta)\frac{\beta^2\sigma^2}{2b} + 2\rho^2\beta^2\eta(1-2\eta\beta) + \eta^2\beta\frac{\sigma^2}{b}.
\end{aligned}
$$

(a) is by using the trick: $\|\nabla L(\boldsymbol{w}_{t-1/2})\|^2 = -\|\nabla L(w_t)\|^2 + \|\nabla L(\boldsymbol{w}_{t-1/2}) - \nabla L(\boldsymbol{w}_t)\|^2 + 2\langle\nabla L(\boldsymbol{w}_{t-1/2}), \nabla L(\boldsymbol{w}_t)\rangle$. (b) is by using Lemma A.6. The last inequality is based on the fact that $\beta^2\mathbb{E}\|\boldsymbol{w}_{t-1/2} - \boldsymbol{w}_t\|^2 \leq \beta^2\mathbb{E}\|(\rho-\eta\|\nabla L(\boldsymbol{w}_t)\|)\frac{\nabla L(\boldsymbol{w}_t)}{\|\nabla L(\boldsymbol{w}_t)\|}\|^2 \leq \beta^2\rho^2 + \eta^2\beta^2\mathbb{E}\|\nabla L(\boldsymbol{w}_t)\|^2$, and the assumption $\eta \leq \frac{1}{2\beta}$. Finally, after simplification, we obtain the result. $\square$

**Theorem 4.5.** *Assume that* $\eta_t = \min\left(\frac{1}{2\beta}, \frac{1}{\sqrt{T}}\right)$, $\rho = \frac{1}{\sqrt{T}}$ *in Algorithm 1, then EG-SAM satisfies* $\frac{1}{T}\sum_{t=0}^{T}\mathbb{E}\|\nabla_{\boldsymbol{w}_t}L(\boldsymbol{w}_t)\|^2 = O\left(\frac{1}{\sqrt{T}} + \frac{1}{\sqrt{T}b}\right)$.

*Proof.* With $\eta_t = \min\left(\frac{1}{2\beta}, \frac{2.99}{4\beta}\right) = \frac{1}{2\beta}$, we have $\left(\eta\beta - \eta\beta^3 - 2\eta\beta\left(\frac{1}{2}+\eta\beta\right) + \left(\frac{1}{2}+\eta\beta\right)\right) > 0$. By further using Lemma A.7 and the above analysis, there exists a positive constant $C$ such that:

$$
\begin{aligned}
C\eta\mathbb{E}\|\nabla L(\boldsymbol{w}_t)\|^2 &\leq \mathbb{E}L(\boldsymbol{w}_t) - \mathbb{E}L(\boldsymbol{w}_{t+1}) + \eta^3(1-2\eta\beta)\frac{\beta^2\sigma^2}{2b} + 2\rho^2\eta(1-2\eta\beta) + \eta^2\beta\frac{\sigma^2}{b} \\
&\quad +\eta^2\beta^3\rho^2 + \eta\beta\rho G.
\end{aligned}
$$

Setting $\rho = \frac{1}{\sqrt{T}}$ and $\eta_t = \min\left(\frac{1}{2\beta}, \frac{1}{\sqrt{T}}\right)$. By telescoping from $t = 1$ to $T$, and divide by $T$,

$$
\begin{aligned}
\frac{C}{T}\sum_{t=1}^{T}\mathbb{E}\|\nabla L(\boldsymbol{w}_t)\|^2 \\
&\leq \frac{L(\boldsymbol{w}_0) - \mathbb{E}L(\boldsymbol{w}_{T+1})}{\eta T} + \eta^2\frac{\beta^2\sigma^2}{2b} + 2\rho^2(1-2\eta\beta) + \eta\beta\frac{\sigma^2}{b} + \eta\beta^3\rho^2 + \beta\rho G \\
&= \frac{L(\boldsymbol{w}_0) - \mathbb{E}L(\boldsymbol{w}_{T+1})}{\sqrt{T}} + \frac{\left(\beta^2\sigma^2 b/\sqrt{T} + 2b/\sqrt{T} + 2\beta\sigma^2 + 2b\beta^2/T + 2\beta G\right)}{2\sqrt{T}b} \\
&= O\left(\frac{1}{\sqrt{T}} + \frac{1}{\sqrt{T}b}\right).
\end{aligned}
$$

$\square$

### A.3 OG-SAM CONVERGENCE

Recall from Section A.2 that $L_t(\boldsymbol{w}) := \frac{1}{b}\sum_{i\in I_t}\ell_i(\boldsymbol{w})$. In the following, we assume $\rho, \eta, \beta > 0$. Also, we use assumptions 4.3, 4.4 and 4.5.

**Lemma A.8.** *The update scheme of OG-SAM is equivalent to*

$$
\hat{\boldsymbol{w}}_t = \hat{\boldsymbol{w}}_{t-1} - 2\eta_t \nabla_{\hat{\boldsymbol{w}}_{t-1}} L \left( \hat{\boldsymbol{w}}_{t-1} + \hat{\boldsymbol{\epsilon}}_{t-1} \Big|_{\hat{\boldsymbol{\epsilon}}_{t-1} = \frac{\rho \nabla_{\boldsymbol{w}_{t-1}} L(\boldsymbol{w}_{t-1})}{\|\nabla_{\hat{\boldsymbol{w}}_{t-1}} L(\boldsymbol{w}_{t-1})\|}} \right)
$$

$$
+ \eta_{t-1} \nabla_{\hat{\boldsymbol{w}}_{t-2}} L \left( \hat{\boldsymbol{w}}_{t-2} + \hat{\boldsymbol{\epsilon}}_{t-2} \Big|_{\hat{\boldsymbol{\epsilon}}_{t-2} = \frac{\rho \nabla_{\boldsymbol{w}_{t-2}} L(\boldsymbol{w}_{t-2})}{\|\nabla_{\boldsymbol{w}_{t-2}} L(\boldsymbol{w}_{t-2})\|}} \right).
$$

*Proof.* Recall that in OG-SAM, we have:

$$
\hat{\boldsymbol{w}}_t = \boldsymbol{w}_{t-1} - \eta_t \nabla_{\hat{\boldsymbol{w}}_{t-1}} L(\hat{\boldsymbol{w}}_{t-1} + \hat{\boldsymbol{\epsilon}}_{t-1}), \tag{21}
$$

and

$$
\boldsymbol{w}_{t-1} = \boldsymbol{w}_{t-2} - \eta_t \nabla_{\hat{\boldsymbol{w}}_t} L(\hat{\boldsymbol{w}}_{t-1} + \hat{\boldsymbol{\epsilon}}_{t-1}). \tag{22}
$$

Substitute $\boldsymbol{w}_{t-1}$ in (21) by (22), we have

$$
\hat{\boldsymbol{w}}_t = \boldsymbol{w}_{t-2} - 2\eta_t \nabla_{\hat{\boldsymbol{w}}_{t-1}} L(\hat{\boldsymbol{w}}_{t-1} + \hat{\boldsymbol{\epsilon}}_{t-1}). \tag{23}
$$

Also, note that in OG-SAM,

$$
\hat{\boldsymbol{w}}_{t-1} + \eta_{t-1} \nabla_{\hat{\boldsymbol{w}}_{t-2}} L(\hat{\boldsymbol{w}}_{t-2} + \hat{\boldsymbol{\epsilon}}_{t-2}) = \boldsymbol{w}_{t-2}. \tag{24}
$$

Substitute (24) into (23), we have:

$$
\hat{\boldsymbol{w}}_t = \hat{\boldsymbol{w}}_{t-1} - 2\eta_t \nabla_{\hat{\boldsymbol{w}}_{t-1}} L(\hat{\boldsymbol{w}}_{t-1} + \hat{\boldsymbol{\epsilon}}_{t-1} | \hat{\boldsymbol{\epsilon}}_{t-1} = \frac{\rho \nabla_{\boldsymbol{w}_{t-1}} L(\boldsymbol{w}_{t-1})}{\|\nabla_{\boldsymbol{w}_{t-1}} L(\boldsymbol{w}_{t-1})\|})
$$

$$
+ \eta_{t-1} \nabla_{\hat{\boldsymbol{w}}_{t-2}} L(\hat{\boldsymbol{w}}_{t-2} + \hat{\boldsymbol{\epsilon}}_{t-2} | \hat{\boldsymbol{\epsilon}}_{t-2} = \frac{\rho \nabla_{\boldsymbol{w}_{t-2}} L(\boldsymbol{w}_{t-2})}{\|\nabla_{\hat{\boldsymbol{w}}_{t-2}} L(\boldsymbol{w}_{t-2})\|}),
$$

which is the desired result. $\square$

**Lemma A.9.**

$$
\left\langle \nabla L \left( \boldsymbol{w} + \rho \frac{\nabla L(\boldsymbol{w})}{\|\nabla L(\boldsymbol{w})\|} \right), \nabla L(\boldsymbol{w}) \right\rangle \geq \|\nabla L(\boldsymbol{w})\|^2 - \beta\rho\|\nabla L(\boldsymbol{w})\|.
$$

*Proof.*

$$
\left\langle \nabla L(\boldsymbol{w} + \rho \frac{\nabla L(\boldsymbol{w})}{\|\nabla L(\boldsymbol{w})\|}), \nabla L(\boldsymbol{w}) \right\rangle
$$

$$
= \left\langle \nabla L \left( \boldsymbol{w} + \rho \frac{\nabla L(\boldsymbol{w})}{\|\nabla L(\boldsymbol{w})\|} \right) - \nabla L(\boldsymbol{w}), \nabla L(\boldsymbol{w}) \right\rangle + \|\nabla L(\boldsymbol{w})\|^2
$$

$$
= \frac{\|\nabla L(\boldsymbol{w})\|}{\rho} \left\langle \nabla L(\boldsymbol{w} + \rho \frac{\nabla L(\boldsymbol{w})}{\|\nabla L(\boldsymbol{w})\|}) - \nabla L(\boldsymbol{w}), \frac{\rho}{\|\nabla L(\boldsymbol{w})\|} \nabla L(\boldsymbol{w}) \right\rangle + \|\nabla L(\boldsymbol{w})\|^2
$$

$$
\overset{(a)}{\geq} \left( 1 - \frac{\beta\rho}{\|\nabla L(\boldsymbol{w})\|} \right) \|\nabla L(\boldsymbol{w})\|^2
$$

$$
\overset{(b)}{\geq} \|\nabla L(\boldsymbol{w})\|^2 - \beta\rho\|\nabla L(\boldsymbol{w})\|
$$

(a) is the co-coercivity property. (b) is by simple calculation. $\square$

**Lemma A.10.**

$$
\mathbb{E}\left[ \left\langle \nabla_{\boldsymbol{w}_{t-1}} [L(\boldsymbol{w}_{t-1})], \nabla_{\boldsymbol{w}_{t-1}} L_t \left( \boldsymbol{w}_{t-1} + \rho \frac{\nabla_{\boldsymbol{w}_{t-1}} L_t(\boldsymbol{w}_{t-1})}{\|\nabla_{\boldsymbol{w}_{t-1}} L_t(\boldsymbol{w}_{t-1})\|} \right) \right\rangle \right]
$$

$$
\geq \frac{1}{2}\|\nabla L(\boldsymbol{w}_{t-1})\|^2 - \beta\rho\|\nabla L(\boldsymbol{w}_{t-1})\| - \rho^2\beta^2.
$$

*Proof.* Similar to the proof of Lemma A.6, note that

$$\left\langle \nabla L_t\left(\boldsymbol{w}_{t-1} + \rho\frac{\nabla L_t(\boldsymbol{w}_{t-1})}{\|\nabla L_t(\boldsymbol{w}_{t-1})\|}\right), \nabla L(\boldsymbol{w}_{t-1})\right\rangle$$

$$= \left\langle \nabla L_t\left(\boldsymbol{w}_{t-1} + \rho\frac{\nabla L_t(\boldsymbol{w}_{t-1})}{\|\nabla L_t(\boldsymbol{w}_{t-1})\|}\right) - \nabla L_t(\boldsymbol{w}_{t-1} + \rho\frac{\nabla L(\boldsymbol{w}_{t-1})}{\|\nabla L(\boldsymbol{w}_{t-1})\|}), \nabla L(\boldsymbol{w}_{t-1})\right\rangle$$

$$- \left\langle -\nabla L_t(\boldsymbol{w}_{t-1} + \rho\frac{\nabla L(\boldsymbol{w}_{t-1})}{\|\nabla L(\boldsymbol{w}_{t-1})\|}), \nabla L(\boldsymbol{w}_{t-1})\right\rangle.$$

To bound the RHS,

$$-\mathbb{E}\left\langle \nabla L_t\left(\boldsymbol{w}_{t-1} + \rho\frac{\nabla L_t(\boldsymbol{w}_{t-1})}{\|\nabla L_t(\boldsymbol{w}_{t-1})\|}\right) - \nabla L_t(\boldsymbol{w}_{t-1} + \rho\frac{\nabla L(\boldsymbol{w}_{t-1})}{\|\nabla L(\boldsymbol{w}_{t-1})\|}), \nabla L(\boldsymbol{w}_{t-1})\right\rangle$$

$$\leq \frac{1}{2}\mathbb{E}\|\nabla L_t\left(\boldsymbol{w}_{t-1} + \rho\frac{\nabla L_t(\boldsymbol{w}_{t-1})}{\|\nabla L_t(\boldsymbol{w}_{t-1})\|}\right) - \nabla L_t\left(\boldsymbol{w}_{t-1} + \rho\frac{\nabla L(\boldsymbol{w}_{t-1})}{\|\nabla L(\boldsymbol{w}_{t-1})\|}\right)\|^2 + \frac{1}{2}\|\nabla L(\boldsymbol{w}_{t-1})\|^2$$

$$\leq \frac{\beta^2}{2}\mathbb{E}\left\|\rho\frac{\nabla L_t(\boldsymbol{w}_{t-1})}{\|\nabla L_t(\boldsymbol{w}_{t-1})\|} - \rho\frac{\nabla L(\boldsymbol{w}_{t-1})}{\|\nabla L(\boldsymbol{w}_{t-1})\|}\right\|^2 + \frac{1}{2}\|\nabla L(\boldsymbol{w}_{t-1})\|^2$$

$$\leq \rho^2\beta^2 + \frac{1}{2}\|\nabla L(\boldsymbol{w}_{t-1})\|^2.$$

The first inequality is by the Young's inequality. Also, it has been proven that

$$\left\langle \nabla L_t(\boldsymbol{w}_{t-1} + \rho\frac{\nabla L(\boldsymbol{w}_{t-1})}{\|\nabla L(\boldsymbol{w}_{t-1})\|}), \nabla L(\boldsymbol{w}_{t-1})\right\rangle \geq \|\nabla L(\boldsymbol{w}_{t-1})\|^2 - \beta\rho\|\nabla L(\boldsymbol{w}_{t-1})\|.$$

Combining the two inequalities together, we obtain the desired result. □

**Lemma A.11.**

$$\mathbb{E}\Big[\Big\langle \nabla_{\hat{\boldsymbol{w}}_{t-1}}L(\hat{\boldsymbol{w}}_{t-1}),$$

$$\nabla_{\hat{\boldsymbol{w}}_{t-2}}\left[L_{t-1}\left(\hat{\boldsymbol{w}}_{t-2} + \rho\frac{\nabla_{\boldsymbol{w}_{t-2}}L_{t-1}(\boldsymbol{w}_{t-2})}{\|\nabla_{\boldsymbol{w}_{t-2}}L_{t-1}(\boldsymbol{w}_{t-2})\|}\right) - \nabla_{\boldsymbol{w}_{t-1}}\left[L_t\left(\boldsymbol{w}_{t-1} + \rho\frac{\nabla_{\hat{\boldsymbol{w}}_{t-1}}L_t(\boldsymbol{w}_{t-1})}{\|\nabla_{\boldsymbol{w}_{t-1}}L_t(\boldsymbol{w}_{t-1})\|}\right)\right]\right]\Big\rangle\Big]$$

$$\leq G\left(\beta^2\rho^2 + \frac{5\beta^2\eta^2 G^2}{2}\right)^{\frac{1}{2}}.$$

*Proof.*

$$\mathbb{E}\left[\left\langle \nabla_{\hat{\boldsymbol{w}}_{t-1}}[L(\hat{\boldsymbol{w}}_{t-1})], \nabla_{\hat{\boldsymbol{w}}_{t-2}}L_{t-1}(\hat{\boldsymbol{w}}_{t-2} + \rho\frac{\nabla_{\boldsymbol{w}_{t-2}}L_{t-1}(\boldsymbol{w}_{t-2})}{\|\nabla_{\boldsymbol{w}_{t-2}}L_{t-1}(\boldsymbol{w}_{t-2})\|}) - \nabla_{\hat{\boldsymbol{w}}_{t-1}}[L_t(\hat{\boldsymbol{w}}_{t-1} + \rho\frac{\nabla_{\boldsymbol{w}_{t-1}}L_t(\boldsymbol{w}_{t-1})}{\|\nabla_{\boldsymbol{w}_{t-1}}L_t(\boldsymbol{w}_{t-1})\|})]\right\rangle\right]$$

$$\leq \left(\mathbb{E}\|\nabla_{\hat{\boldsymbol{w}}_{t-1}}[L(\hat{\boldsymbol{w}}_{t-1})]\|^2\right)^{\frac{1}{2}}\left(\mathbb{E}\|\nabla_{\hat{\boldsymbol{w}}_{t-2}}L_{t-1}(\hat{\boldsymbol{w}}_{t-2} + \rho\frac{\nabla_{\boldsymbol{w}_{t-2}}L_{t-1}(\boldsymbol{w}_{t-2})}{\|\nabla_{\boldsymbol{w}_{t-2}}L_{t-1}(\boldsymbol{w}_{t-2})\|}) - \nabla_{\hat{\boldsymbol{w}}_{t-1}}[L_t(\hat{\boldsymbol{w}}_{t-1} + \rho\frac{\nabla_{\boldsymbol{w}_{t-1}}L_t(\boldsymbol{w}_{t-1})}{\|\nabla_{\boldsymbol{w}_{t-1}}L_t(\boldsymbol{w}_{t-1})\|})]\|^2\right)^{\frac{1}{2}}$$

$$\leq G\left(\left(\beta^2\rho^2 + \frac{\beta^2}{2}\mathbb{E}\left[\|2\eta\nabla_{\hat{\boldsymbol{w}}_{t-2}}[L_{t-1}(\hat{\boldsymbol{w}}_{t-2} + \rho\frac{\nabla_{\boldsymbol{w}_{t-2}}L_{t-1}(\boldsymbol{w}_{t-2})}{\|\nabla_{\boldsymbol{w}_{t-2}}L_{t-1}(\boldsymbol{w}_{t-2})\|}) - \eta\nabla_{\hat{\boldsymbol{w}}_{t-3}}[L_{t-2}(\hat{\boldsymbol{w}}_{t-3} + \rho\frac{\nabla_{\boldsymbol{w}_{t-3}}L_{t-2}(\boldsymbol{w}_{t-3})}{\|\nabla_{\boldsymbol{w}_{t-3}}L_{t-2}(\boldsymbol{w}_{t-3})\|})\|^2]\right)^2\right)^{\frac{1}{2}}$$

$$\leq G\left(\beta^2\rho^2 + \frac{5\beta^2\eta^2 G^2}{2}\right)^{\frac{1}{2}}.$$

The first inequality uses the Cauchy-Schwartz inequality. The last inequality uses the triangle inequality. □

**Lemma A.12.** *With assumptions 4.3, 4.4 and 4.5,*

$$\mathbb{E}\left\|2\eta\nabla_{\hat{\boldsymbol{w}}_{t-1}}L_t(\hat{\boldsymbol{w}}_{t-1} + \rho\frac{\nabla_{\boldsymbol{w}_{t-1}}L_t(\boldsymbol{w}_{t-1})}{\|\nabla_{\boldsymbol{w}_{t-1}}L_t(\boldsymbol{w}_{t-1})\|}) - \eta\nabla_{\hat{\boldsymbol{w}}_{t-2}}L_{t-1}(\hat{\boldsymbol{w}}_{t-2} + \rho\frac{\nabla_{\boldsymbol{w}_{t-2}}L_{t-1}(\boldsymbol{w}_{t-2})}{\|\nabla_{\boldsymbol{w}_{t-2}}L_{t-1}(\boldsymbol{w}_{t-2})\|})\right\|^2$$

$$\leq \frac{4\eta^2\beta^2\rho^2}{b} + \frac{4\eta^2\sigma^2}{b} + \eta^2 G^2.$$

*Proof.*

$$\mathbb{E}\left\|2\eta\nabla_{\hat{\boldsymbol{w}}_{t-1}}L_t(\hat{\boldsymbol{w}}_{t-1}+\rho\frac{\nabla_{\boldsymbol{w}_{t-1}}L_t(\boldsymbol{w}_{t-1})}{\|\nabla_{\boldsymbol{w}_{t-1}}L_t(\boldsymbol{w}_{t-1})\|})-\eta\nabla_{\hat{\boldsymbol{w}}_{t-2}}L_{t-1}(\hat{\boldsymbol{w}}_{t-2}+\rho\frac{\nabla_{\boldsymbol{w}_{t-2}}L_{t-1}(\boldsymbol{w}_{t-2})}{\|\nabla_{\boldsymbol{w}_{t-2}}L_{t-1}(\boldsymbol{w}_{t-2})\|})\right\|^2$$

$$\leq \quad \mathbb{E}\|2\eta\nabla_{\hat{\boldsymbol{w}}_{t-1}}L_t(\hat{\boldsymbol{w}}_{t-1}+\rho\frac{\nabla_{\boldsymbol{w}_{t-1}}L_t(\boldsymbol{w}_{t-1})}{\|\nabla_{\boldsymbol{w}_{t-1}}L_t(\boldsymbol{w}_{t-1})\|})\|^2 + \mathbb{E}\|\eta\nabla_{\hat{\boldsymbol{w}}_{t-2}}L_{t-1}(\hat{\boldsymbol{w}}_{t-2}+\rho\frac{\nabla_{\boldsymbol{w}_{t-2}}L_{t-1}(\boldsymbol{w}_{t-2})}{\|\nabla_{\boldsymbol{w}_{t-2}}L_{t-1}(\boldsymbol{w}_{t-2})\|})\|^2$$

$$\leq \quad 4\eta^2\mathbb{E}\left\|\nabla_{\hat{\boldsymbol{w}}_{t-1}}L_t(\hat{\boldsymbol{w}}_{t-1}+\rho\frac{\nabla_{\boldsymbol{w}_{t-1}}L_t(\boldsymbol{w}_{t-1})}{\|\nabla_{\boldsymbol{w}_{t-1}}L_t(\hat{\boldsymbol{w}}_{t-1})\|})-\nabla_{\boldsymbol{w}_{t-1}}L_t(\hat{\boldsymbol{w}}_{t-1})\right\|^2 + \eta^2\mathbb{E}\|\nabla_{\hat{\boldsymbol{w}}_{t-1}}L_t(\hat{\boldsymbol{w}}_{t-1})-\nabla_{\hat{\boldsymbol{w}}_{t-1}}L(\hat{\boldsymbol{w}}_{t-1})\|^2$$

$$+\eta^2 G^2$$

$$\leq \quad \frac{4\eta^2\beta^2\mathbb{E}\left[\|\rho\frac{\nabla_{\boldsymbol{w}_{t-1}}L_t(\boldsymbol{w}_{t-1})}{\|\nabla_{\boldsymbol{w}_{t-1}}L_t(\boldsymbol{w}_{t-1})\|}\|^2\right]}{b} + \frac{4\eta^2\sigma^2}{b} + \eta^2 G^2$$

$$\leq \quad \frac{4\eta^2\beta^2\rho^2}{b} + \frac{4\eta^2\sigma^2}{b} + \eta^2 G^2.$$

$\square$

**Lemma A.13.** *If $\rho < \frac{1}{2\beta}$,*

$$\mathbb{E}[L(\boldsymbol{w}_{t-1})] \leq \mathbb{E}[L(\hat{\boldsymbol{w}}_t)] + \frac{\beta\eta_T^2}{2}G^2.$$

*Proof.* Based on the update scheme of OG-SAM, $\boldsymbol{w}_{t-1} = \hat{\boldsymbol{w}}_t + \eta_t\mathbf{g}_{t-1}$. By the smoothness assumption on Assumption 4.4,

$$L(\boldsymbol{w}_{t-1}) - L(\hat{\boldsymbol{w}}_t) - \eta_t\langle\nabla L(\hat{\boldsymbol{w}}_t),\nabla L_t(\boldsymbol{w}_t)\rangle \leq \frac{\beta\eta_t^2}{2}\|\mathbf{g}_{t-1}\|^2 + \eta_t G^2.$$

Also, based on the Young's inequality and assumption 4.5,

$$\mathbb{E}\left[\langle\nabla L(\boldsymbol{w}_{t-1}+\eta_t\mathbf{g}_{t-1}),\nabla L_t(\boldsymbol{w}_{t-1})\rangle\right] \leq G^2$$

Therefore,

$$\mathbb{E}[L(\boldsymbol{w}_{t-1})] \leq \mathbb{E}[L(\hat{\boldsymbol{w}}_t)] + \frac{\beta\eta_t^2}{2}\|\mathbf{g}_{t-1}\|^2 + \eta_t G^2.$$

$\square$

**Lemma A.14.** *With assumptions 4.3, 4.4 and 4.5, we have:*

$$\frac{1}{2}\eta\mathbb{E}\|\nabla L(\hat{\boldsymbol{w}}_{t-1})\|^2 \quad \leq \quad \mathbb{E}[L(\hat{\boldsymbol{w}}_{t-1})] - \mathbb{E}[L(\hat{\boldsymbol{w}}_t)] + \eta\beta\rho E\|\nabla L(\hat{\boldsymbol{w}}_t)\| + \eta\rho^2\beta^2$$

$$+\eta\left[G\left(\beta^2\rho^2+\frac{5\beta^2\eta^2G^2}{2}\right)^{\frac{1}{2}}\right] + \frac{\beta}{2}\left(\frac{4\eta^2\beta^2\rho^2}{b}+4\eta^2\sigma^2+\eta^2G^2\right).$$

*Proof.* Using the definition of smoothness on Assumption (4.4) and taking the expectation on both sides, we have:

$$\mathbb{E}[L(\hat{\boldsymbol{w}}_t)] \leq \mathbb{E}[L(\hat{\boldsymbol{w}}_{t-1})]$$

$$-\eta\mathbb{E}\left[\left\langle\nabla_{\hat{\boldsymbol{w}}_{t-1}}[L(\hat{\boldsymbol{w}}_{t-1})],2\nabla_{\hat{\boldsymbol{w}}_{t-1}}[L_t(\hat{\boldsymbol{w}}_{t-1}+\rho\frac{\nabla_{\boldsymbol{w}_{t-1}}L_t(\boldsymbol{w}_{t-1})}{\|\nabla_{\boldsymbol{w}_{t-1}}L_t(\boldsymbol{w}_{t-1})\|})-\nabla_{\hat{\boldsymbol{w}}_{t-2}}[L_{t-1}(\boldsymbol{w}_{t-2}+\rho\frac{\nabla_{\boldsymbol{w}_{t-2}}L_{t-1}(\boldsymbol{w}_{t-2})}{\|\nabla_{\boldsymbol{w}_{t-2}}L_{t-1}(\boldsymbol{w}_{t-2})\|})\right\rangle\right]$$

$$+\frac{\beta}{2}\mathbb{E}\left\|2\eta\nabla_{\hat{\boldsymbol{w}}_{t-1}}[L_t(\hat{\boldsymbol{w}}_{t-1}+\rho\frac{\nabla_{\boldsymbol{w}_{t-1}}L_t(\boldsymbol{w}_{t-1})}{\|\nabla_{\boldsymbol{w}_{t-1}}L_t(\boldsymbol{w}_{t-1})\|})-\eta\nabla_{\hat{\boldsymbol{w}}_{t-2}}[L_{t-1}(\hat{\boldsymbol{w}}_{t-2}+\rho\frac{\nabla_{\boldsymbol{w}_{t-2}}L_{t-1}(\boldsymbol{w}_{t-2})}{\|\nabla_{\boldsymbol{w}_{t-2}}L_{t-1}(\boldsymbol{w}_{t-2})\|})\right\|^2$$

$$\leq \quad \mathbb{E}[L(\hat{\boldsymbol{w}}_{t-1})] - \eta E\left[\left\langle\nabla_{\hat{\boldsymbol{w}}_{t-1}}[L(\hat{\boldsymbol{w}}_{t-1})],\nabla_{\hat{\boldsymbol{w}}_{t-1}}[L_{t-1}(\hat{\boldsymbol{w}}_{t-1}+\rho\frac{\nabla_{\boldsymbol{w}_{t-1}}L_t(\boldsymbol{w}_{t-1})}{\|\nabla_{\boldsymbol{w}_{t-1}}L_t(\boldsymbol{w}_{t-1})\|})\right\rangle\right]$$

$$+\eta\mathbb{E}\left[\left\langle\nabla_{\hat{\boldsymbol{w}}_{t-1}}[L(\hat{\boldsymbol{w}}_{t-1})],\nabla_{\hat{\boldsymbol{w}}_{t-2}}[L_{t-1}(\hat{\boldsymbol{w}}_{t-2}+\rho\frac{\nabla_{\boldsymbol{w}_{t-2}}L_t(\boldsymbol{w}_{t-2})}{\|\nabla_{\boldsymbol{w}_{t-2}}L_{t-1}(\boldsymbol{w}_{t-2})\|})-\nabla_{\hat{\boldsymbol{w}}_{t-1}}[L_t(\boldsymbol{w}_{t-1}+\rho\frac{\nabla_{\boldsymbol{w}_{t-1}}L_t(\boldsymbol{w}_{t-1})}{\|\nabla_{\boldsymbol{w}_{t-1}}L_t(\hat{\boldsymbol{w}}_{t-1})\|})]\right\rangle\right]$$

$$+\frac{\eta^2\beta}{2}\mathbb{E}\left\|2\nabla_{\hat{\boldsymbol{w}}_{t-1}}[L_t(\hat{\boldsymbol{w}}_{t-1}+\rho\frac{\nabla_{\boldsymbol{w}_{t-1}}L_t(\boldsymbol{w}_{t-1})}{\|\nabla_{\boldsymbol{w}_{t-1}}L_t(\boldsymbol{w}_{t-1})\|})-\nabla_{\hat{\boldsymbol{w}}_{t-2}}[L_{t-1}(\hat{\boldsymbol{w}}_{t-2}+\rho\frac{\nabla_{\boldsymbol{w}_{t-2}}L_{t-1}(\boldsymbol{w}_{t-2})}{\|\nabla_{\boldsymbol{w}_{t-2}}L_{t-1}(\boldsymbol{w}_{t-2})\|})\right\|^2.$$

Using Lemmas A.12, A.11 and A.10 above,

$$
\begin{aligned}
\mathbb{E}[L(\hat{\boldsymbol{w}}_t)] &\leq \mathbb{E}[L(\hat{\boldsymbol{w}}_{t-1})] - \eta \mathbb{E}\left[\left\langle \nabla_{\hat{\boldsymbol{w}}_{t-1}}[L(\hat{\boldsymbol{w}}_{t-1})], \nabla_{\hat{\boldsymbol{w}}_{t-1}}[L_t(\hat{\boldsymbol{w}}_{t-1} + \rho \frac{\nabla_{\boldsymbol{w}_{t-1}} L_t(\boldsymbol{w}_{t-1})}{\|\nabla_{\boldsymbol{w}_{t-1}} L_t(\boldsymbol{w}_{t-1})\|})\right\rangle\right] \\
&+ \eta \mathbb{E}\left[\left\langle \nabla_{\hat{\boldsymbol{w}}_{t-1}}[L(\hat{\boldsymbol{w}}_{t-1})], \nabla_{\hat{\boldsymbol{w}}_{t-2}}[L_{t-1}(\hat{\boldsymbol{w}}_{t-2} + \rho \frac{\nabla_{\boldsymbol{w}_{t-2}} L_{t-1}(\boldsymbol{w}_{t-2})}{\|\nabla_{\boldsymbol{w}_{t-2}} L_{t-1}(\boldsymbol{w}_{t-2})\|}) - \nabla_{\hat{\boldsymbol{w}}_{t-1}}[L_t(\hat{\boldsymbol{w}}_{t-1} + \rho \frac{\nabla_{\boldsymbol{w}_{t-1}} L_t(\boldsymbol{w}_{t-1})}{\|\nabla_{\boldsymbol{w}_{t-1}} L_t(\boldsymbol{w}_{t-1})\|})]\right\rangle\right] \\
&+ \frac{\beta}{2}\mathbb{E}\left\|2\eta\nabla_{\hat{\boldsymbol{w}}_{t-1}}[L_t(\hat{\boldsymbol{w}}_{t-1} + \rho \frac{\nabla_{\boldsymbol{w}_{t-1}} L_t(\boldsymbol{w}_{t-1})}{\|\nabla_{\boldsymbol{w}_{t-1}} L_t(\boldsymbol{w}_{t-1})\|}) - \eta\nabla_{\hat{\boldsymbol{w}}_{t-2}}[L_{t-1}(\hat{\boldsymbol{w}}_{t-2} + \rho \frac{\nabla_{\boldsymbol{w}_{t-2}} L_{t-1}(\boldsymbol{w}_{t-2})}{\|\nabla_{\boldsymbol{w}_{t-2}} L_{t-1}(\boldsymbol{w}_{t-2})\|})\right\|^2 \\
&\leq \mathbb{E}[L(\hat{\boldsymbol{w}}_{t-1})] - \eta\mathbb{E}\left[\|\nabla L(\hat{\boldsymbol{w}}_{t-1})\|^2 - \beta\rho\|\nabla L(\hat{\boldsymbol{w}}_{t-1})\| - \rho^2\beta^2\right] \\
&+ \eta\mathbb{E}\left[G\left(\beta^2\rho^2 + \frac{5\beta^2\eta^2 G^2}{2}\right)^{\frac{1}{2}}\right] + \frac{\beta\eta^2}{2}\left(\frac{4\beta^2\rho^2}{b} + 4\sigma^2 + G^2\right).
\end{aligned}
$$

This can then be simplified as:

$$
\begin{aligned}
\frac{1}{2}\eta\mathbb{E}\|\nabla L(\hat{\boldsymbol{w}}_{t-1})\|^2 &\leq \mathbb{E}[L(\hat{\boldsymbol{w}}_{t-1})] - \mathbb{E}[L(\hat{\boldsymbol{w}}_t)] + \eta\beta\rho\mathbb{E}\|\nabla L(\hat{\boldsymbol{w}}_t)\| + \eta\rho^2\beta^2 \\
&+ \eta\left[G\left(\beta^2\rho^2 + \frac{5\beta^2\eta^2 G^2}{2}\right)^{\frac{1}{2}}\right] + \frac{\beta}{2}\left(\frac{4\eta^2\beta^2\rho^2}{b} + 4\eta^2\sigma^2 + \eta^2 G^2\right),
\end{aligned}
$$

which is the desired result. $\qquad\square$

**Theorem 4.6.** *Assume that* $\rho_t = \min\left(\frac{1}{\sqrt{T}}, \frac{1}{\beta}\right)$, $\eta_t = \min\left(\frac{1}{\sqrt{T}}, \frac{1}{\beta}\right)$ *in Algorithm 2, then OG-SAM satisfies:* $\frac{1}{T}\sum_{t=0}^T \mathbb{E}\|\nabla_{\boldsymbol{w}_t} L(\boldsymbol{w}_t)\|^2 = O\left(\frac{1}{\sqrt{T}} + \frac{1}{\sqrt{T}b}\right).$

*Proof.* Using Lemma A.14, and with $\rho_t = \min\left(\frac{1}{\sqrt{T}}, \frac{1}{\beta}\right)$, $\eta_t = \min\left(\frac{1}{\sqrt{T}}, \frac{1}{\beta}\right)$, and the fact that $\sqrt{a+b} \leq \sqrt{a} + \sqrt{b}$, we have

$$
\begin{aligned}
\mathbb{E}\|\nabla L(\hat{\boldsymbol{w}}_{t-1})\|^2 &\leq \frac{2[\mathbb{E}[L(\hat{\boldsymbol{w}}_{t-1})] - \mathbb{E}[L(\hat{\boldsymbol{w}}_t)]}{\eta} + 2\beta\rho G + 2\rho^2\beta^2 \\
&2\left[\beta\rho + \frac{5\beta\eta G}{2}\right] + 2\eta\beta\left(\frac{4\beta^2\rho^2}{b} + 4\sigma^2 + G^2\right).
\end{aligned}
$$

Telescoping from 1 to $T$, and substitute $\eta_T$ to $\frac{1}{\sqrt{T}}$, we have:

$$
\begin{aligned}
\frac{1}{T}\sum_{t=1}^T \mathbb{E}\|\nabla L(\hat{\boldsymbol{w}}_t)\|^2 &\leq \frac{2(\mathbb{E}[L(\hat{\boldsymbol{w}}_0)] - \mathbb{E}[L(\hat{\boldsymbol{w}}_{t+1})])}{\sqrt{T}\eta} + \frac{2\beta G}{\sqrt{T}} + \frac{2\rho^2\beta^2}{T} \\
&+ \frac{2\beta}{\sqrt{T}} + \frac{5\beta G}{\sqrt{T}} + \frac{2\beta\left(\frac{4\beta^2\rho^2}{b} + 4\sigma^2 + G^2\right)}{T} \\
&\leq O\left(\frac{1}{\sqrt{T}} + \frac{1}{\sqrt{T}b}\right).
\end{aligned}
$$

We obtain:

$$
\frac{1}{T}\sum_{t=0}^T \mathbb{E}\|\nabla_{\hat{\boldsymbol{w}}_t} L(\hat{\boldsymbol{w}}_t)\|^2 = O\left(\frac{1}{\sqrt{T}} + \frac{1}{\sqrt{T}b}\right).
$$

Finally, using Lemma A.13, we have:

$$
\frac{1}{T}\sum_{t=0}^T \mathbb{E}\|\nabla_{\boldsymbol{w}_t} L(\boldsymbol{w}_t)\|^2 \leq \frac{1}{T}\sum_{t=0}^T \mathbb{E}\|\nabla_{\hat{\boldsymbol{w}}_t} L(\hat{\boldsymbol{w}}_t)\|^2 + \frac{\beta + \sqrt{T}G^2}{T} = O\left(\frac{1}{\sqrt{T}} + \frac{1}{\sqrt{T}b}\right).
$$

$\qquad\square$

### A.4 AO-SAM CONVERGENCE

**Theorem 4.7.** *Assume that $\rho_t = \min\left(\frac{1}{\sqrt{T}}, \frac{1}{\beta}\right)$, $\eta_t = \min\left(\frac{1}{\sqrt{T}}, \frac{1}{2\beta}\right)$ in Algorithm 3, then AO-SAM satisfies:*

$$\frac{1}{T}\sum_{t=0}^{T} E\|\nabla_{\boldsymbol{w}_t} L(\boldsymbol{w}_t)\|^2 = O\left(\frac{1}{\sqrt{T}} + \frac{1}{\sqrt{T}b}\right).$$

*Proof.* Note that for AO-SAM,

1. If $\|\frac{1}{b}\sum_{i \in I_t} \nabla_{\boldsymbol{w}_t} \ell_i(\boldsymbol{w}_t)\|^2 < \mu_t + c_t\sigma_t$, then it is the SGD update scheme on epoch $t$.

2. If $\mathbf{g}_{t-1}$ in step 6 of Algorithm 3 is the gradient obtained by OG-SAM, and $\|\frac{1}{b}\sum_{i \in I_t} \nabla_{\boldsymbol{w}_t} \ell_i(\boldsymbol{w}_t)\|^2 \geq \mu_t + c_t\sigma_t$, i.e., $g_{t-1} = \nabla_{\hat{\boldsymbol{w}}_{t-1}}\left[\frac{1}{b}\sum_{i \in I_{t-1}} \ell_i(\hat{\boldsymbol{w}}_{t-1} + \hat{\boldsymbol{\epsilon}}_{t-1})\right]$, then it is the OG-SAM update scheme on epoch $t$.

3. If $\mathbf{g}_{t-1}$ in step 6 of Algorithm 3 is the gradient obtained by SGD, and $\|\frac{1}{b}\sum_{i \in I_t} \nabla_{\boldsymbol{w}_t} \ell_i(\boldsymbol{w}_t)\|^2 \geq \mu_t + c_t\sigma_t$, then it is EG-SAM update scheme on epoch $t$. The reason is that when the last step is SGD, $\mathbf{g}_{t-1} = \nabla_{\boldsymbol{w}_{t-1}} L_t(\boldsymbol{w}_{t-1})$, by following steps 7-8 in Algorithm 3, Therefore,

$$\begin{aligned}
\boldsymbol{w}_t &= \boldsymbol{w}_{t-1} - \eta_t \mathbf{g}_t \\
&= \boldsymbol{w}_{t-1} - \eta_t \nabla_{\hat{\boldsymbol{w}}_t} \frac{1}{b}\sum_{i \in I_t} \ell_i \left(\boldsymbol{w}_{t-1} - \eta_t \nabla_{\boldsymbol{w}_{t-1}} L_t(\boldsymbol{w}_{t-1}) + \frac{\rho \nabla_{\boldsymbol{w}_{t-1}} \frac{1}{b}\sum_{i \in I_t} \ell_i(\boldsymbol{w}_{t-1})}{\left\|\nabla_{\boldsymbol{w}_{t-1}} \frac{1}{b}\sum_{i \in I_t} \ell_i(\boldsymbol{w}_{t-1})\right\|}\right),
\end{aligned}$$

which is exactly EG-SAM.

Therefore, our main goal is to combine these three different schemes together. Define $\zeta_t^1 \in \{0, 1\}$, $\zeta_t^2 \in \{0, 1\}$ and $\zeta_t^3 \in \{0, 1\}$ to indicate which update scheme is used in epoch $t$: $\zeta_t^1 = 1$ means that OG-SAM is used; $\zeta_t^2 = 1$ means that SGD is used; while $\zeta_t^3 = 1$ means that EG-SAM is used. Obviously, $\zeta_t^1 + \zeta_t^2 + \zeta_t^3 = 1$.

Recall that in Theorem 4.5, we have:

$$\begin{aligned}
C\mathbb{E}\|\nabla L(\boldsymbol{w}_t)\|^2 &\leq \mathbb{E}L(\boldsymbol{w}_t) - \mathbb{E}L(\boldsymbol{w}_{t+1}) + \eta^2(1 - 2\eta\beta)\frac{\beta^2\sigma^2}{2} + 2\rho^2\eta(1 - 2\eta\beta) + \eta^2\beta\frac{\sigma^2}{b} \\
&\quad + \eta^2\beta^3\rho^2 + \eta\beta\rho G.
\end{aligned}$$

Recall that in Theorem 4.6, we have:

$$\begin{aligned}
E\|\nabla L(\boldsymbol{w}_{t-1})\|^2 &\leq \frac{2[E[L(\boldsymbol{w}_{t-1})] - E[L(\boldsymbol{w}_t)]]}{\eta} + 2\beta\rho G + 2\rho^2\beta^2 \\
&\quad 2\left[\beta\rho + \frac{5\beta\eta G}{2}\right] + 2\eta\beta\left(\frac{4\beta^2\rho^2}{b} + 4\sigma^2 + G^2\right) + \frac{\beta\eta_t^2}{2}\|\mathbf{g}_{t-1}\|^2 + \eta_t G^2.
\end{aligned}$$

Also from (2.9) in Theorem 2.1 of (Ghadimi & Lan, 2013),

$$\left(\eta_t - \frac{L}{2}\eta_t^2\right)\|\nabla L(\boldsymbol{w}_{t-1})\|^2 \leq L(\boldsymbol{w}_{t-1}) - L(\boldsymbol{w}_t) + \frac{\beta}{2}\eta_t^2\sigma^2,$$

for SGD. By simple calculation and using assumption 4.5, we have

$$\|\nabla L(\boldsymbol{w}_{t-1})\|^2 \leq \frac{E[L(\boldsymbol{w}_{t-1})] - E[L(\boldsymbol{w}_t)]}{\eta_t} + \frac{\beta}{2}\eta_t\sigma^2 + \frac{\beta}{2}\eta_t G^2.$$

Therefore,

$$\mathbb{E}\left[E\|\nabla L(\boldsymbol{w}_{t-1})\|^2\right]$$

$$\leq \zeta_t^1 \left[\frac{E[L(\boldsymbol{w}_{t-1})] - E[L(\boldsymbol{w}_t)]}{\eta} + 2\beta\rho G + 2\rho^2\beta^2 + 2\left[\beta\rho + \frac{5\beta\eta G}{2}\right] + 2\eta\beta\left(\frac{4\beta^2\rho^2}{b} + 4\sigma^2 + G^2\right) + \frac{\beta\eta_t^2}{2}\|\mathbf{g}_{t-1}\|^2 + \eta_t G^2\right]$$

$$+ \zeta_t^2 \left[\frac{E[L(\boldsymbol{w}_{t-1})] - E[L(\boldsymbol{w}_t)]}{\eta} + \frac{\beta}{2}\eta\sigma^2 + \frac{\beta}{2}\eta G^2\right]$$

$$+ \frac{\zeta_t^3}{C}\left[E[L(\boldsymbol{w}_{t-1})] - E[L(\boldsymbol{w}_t)] + \eta(1 - 2\eta\beta)\frac{\beta^2\sigma^2}{2} + 2\rho^2\eta^2(1 - 2\eta\beta) + \eta^2\beta\frac{\sigma^2}{b} + \eta^2\beta^3\rho^2 + G\beta\rho\eta\right]$$

$$\leq \left[\frac{E[L(\boldsymbol{w}_{t-1})] - E[L(\boldsymbol{w}_t)]}{\eta} + 2\beta\rho G + 2\rho^2\beta^2 + 2\left(\beta\rho + \frac{5\beta\eta G}{2}\right) + 2\eta\beta\left(\frac{4\beta^2\rho^2}{b} + 4\sigma^2 + G^2\right) + \frac{\beta\eta_t^2}{2}\|\mathbf{g}_{t-1}\|^2 + \eta_t G^2\right]$$

$$+ \left[\frac{E[L(\boldsymbol{w}_{t-1})] - E[L(\boldsymbol{w}_t)]}{\eta} + \frac{\beta}{2}\eta\sigma^2 + \frac{\beta}{2}\eta G^2\right]$$

$$+ \frac{1}{C}\left[E[L(\boldsymbol{w}_{t-1})] - E[L(\boldsymbol{w}_t)] + \eta^2(1 - 2\eta\beta)\frac{\beta^2\sigma^2}{2} + 2\rho^2\eta(1 - 2\eta\beta) + \eta^2\beta\frac{\sigma^2}{b} + \eta^2\beta^3\rho^2 + G\beta\rho\eta\right].$$

The second equation is due to the fact that $\zeta_1, \zeta_2, \zeta_3 \leq 1$. Note that the summation of RHS from $t = 0$ to $T$ is exactly the summation of EG-SAM, OG-SAM, and SGD together, which has been shown to have $O(\frac{1}{\sqrt{T}} + \frac{1}{\sqrt{T}b})$, $O(\frac{1}{\sqrt{T}} + \frac{1}{\sqrt{T}b})$, and $O(\frac{1}{\sqrt{T}})$ rates in Theorem 4.6, Theorem 4.5, and Theorem 2.1 in (Ghadimi & Lan, 2013), respectively. As the finite summation of $O(\frac{1}{\sqrt{T}} + \frac{1}{\sqrt{T}b})$ is still $O(\frac{1}{\sqrt{T}} + \frac{1}{\sqrt{T}b})$, the result follows. □

### A.5 CONNECTION TO MINIMAX GAME

To further explore the relationship between the converged point of our method and the original objective in SAM (referred to as equation 1), we first introduce the concept of a stationary point in a minimax game, following the definition provided by (Lin et al., 2020b):

Consider a minimax problem $\min_x \max_y f(x, y)$. The stationary point $x$ of this minimax problem satisfies $\|\nabla_x \phi(x)\|^2 = 0$, where $\phi(x) := \max_y f(x, y)$.

Therefore, the stationary point $\boldsymbol{w}$ of SAM (as defined in (1)) is characterized by the condition $\|\nabla_{\boldsymbol{w}} \max_{\boldsymbol{\epsilon}:\|\boldsymbol{\epsilon}\| \leq \rho} L(\boldsymbol{w} + \boldsymbol{\epsilon})\| = 0$. This condition defines the point at which the gradient of the maximized loss with respect to $\boldsymbol{w}$ is zero, indicating a stationary point in the context of the SAM minimax problem.

**Proposition A.15.** *If* $\boldsymbol{w}_t^* \in \{\boldsymbol{w}_t \mid \mathbb{E}\|\nabla L(\boldsymbol{w}_t)\|^2 = 0\}$, *then* $\mathbb{E}\|\nabla \max_{\boldsymbol{\epsilon},\|\boldsymbol{\epsilon}\| \leq \rho} L(\boldsymbol{w}_t^* + \boldsymbol{\epsilon})\|^2 \leq \beta^2\rho^2$.

*Proof.* Note that

$$\mathbb{E}\|\nabla \max_{\boldsymbol{\epsilon},\|\boldsymbol{\epsilon}\| \leq \rho} L(\boldsymbol{w}_t^* + \boldsymbol{\epsilon})\|^2 = \mathbb{E}\|\nabla \max_{\boldsymbol{\epsilon},\|\boldsymbol{\epsilon}\| \leq \rho} L(\boldsymbol{w}_t^* + \boldsymbol{\epsilon}) - \nabla L(\boldsymbol{w}_t^*) + \nabla L(\boldsymbol{w}_t^*)\|^2$$

$$\leq \mathbb{E}\|\nabla \max_{\boldsymbol{\epsilon},\|\boldsymbol{\epsilon}\| \leq \rho} L(\boldsymbol{w}_t^* + \boldsymbol{\epsilon}) - \nabla L(\boldsymbol{w}_t^*)\|^2 + \mathbb{E}\|\nabla L(\boldsymbol{w}_t^*)\|^2$$

$$\overset{(a)}{\leq} \beta^2\rho^2 + 0$$

where the first inequality uses the triangle inequality, and (a) uses the fact that $\mathbb{E}\|\nabla L(\boldsymbol{w}_t)\|^2 = 0$ as well as the $\beta$-smoothness assumption. □

Note that $\{\boldsymbol{w}_t \mid \mathbb{E}\|\nabla L(\boldsymbol{w}_t)\|^2 = 0\}$ is the EG-SAM's stationary point set. The result reveals that the stationary point $\boldsymbol{w}_t^*$ found by EG-SAM is very close to the stationary point of (1), especially when $\rho$ is small.

## B EXTRA EXPERIMENTS

Extra experiments include testing loss, testing accuracy, and training loss for *CIFAR-10* and *CIFAR-100* ( Figure 4); Testing accuracy with noise label on *CIFAR-100* (Figure 6); as well as training and testing accuracy on *CIFAR-10* and *CIFAR-100* (Table 7 ). Fig. 5 reveals the gradient norm.

For Table 7, Besides *ResNet-18* and *WideResNet-28-10*, we also compare our baselines with *PyramidNet-110* (Han et al., 2017), as well as ViT-S16(Dosovitskiy et al., 2021). The number of training epochs in *PyramidNet* is 300, other settings are similar to *ResNet-18*.

All discussion of these experiments are illustrated in Section 5.

Besides, we added a experiment regarding increasing $\rho$ in Table 9. As can been from the table, i.e., $\rho_t = \rho_0 + \frac{t(\rho_T - \rho_0)}{T}$, where $\rho_T$ is the final value $\rho_T$ (1.0 in our experiment), while $\rho_0$ is the initial value of $\rho$ (0.1 in our experiment). our method with increasing $\rho$ still outperforms the SAM, and is comparable to EG-SAM, while SAM with increasing $\rho$ perform worse than original SAM method.

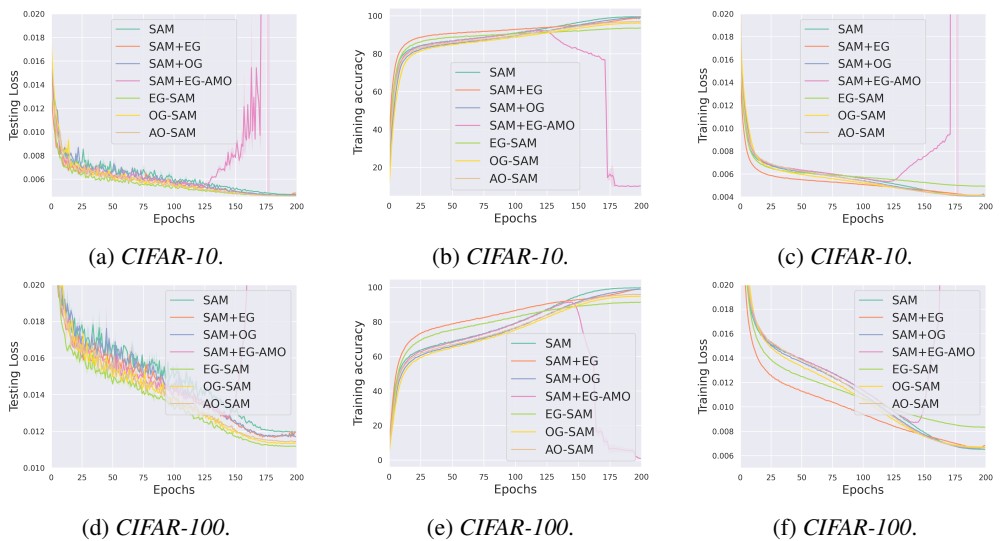

| (a) *CIFAR-10*. | (b) *CIFAR-10*. | (c) *CIFAR-10*. |
|---|---|---|
| (d) *CIFAR-100*. | (e) *CIFAR-100*. | (f) *CIFAR-100*. |

Figure 4: Training accuracy, training and testing losses on *CIFAR-10* and *CIFAR-100* (with ResNet-18 backbone).

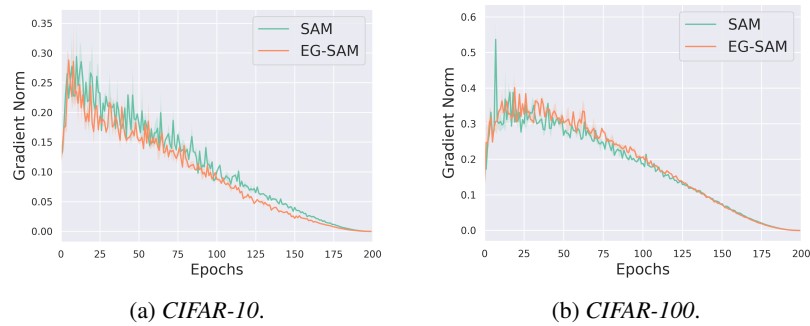

(a) *CIFAR-10*.      (b) *CIFAR-100*.

Figure 5: Gradient Norm on *CIFAR-10* and *CIFAR-100* (with *ResNet-18* backbone).

We also added the performance comparison between GSAM, MSAM, and our AO-SAM in CIFAR-10 and CIFAR-100 with WideResNet-28-10 backbone. From the below table 10, we find that our method also outperforms GSAM and MSAM.

We also provide clock computational time. Table 11 is the clock computational time per epoch on CIFAR-10 with Resnet-18 as the backbone. As can be seen, the higher % SAM ratio means more clock computational time, and our proposed OG-SAM has relatively same training time as SAM, while AO-SAM is faster than SAM.

Table 6: Testing accuracy and fraction of SAM updates on *CIFAR-100* with different levels of label noise. Methods are grouped based on %SAM. The best accuracy is in bold. The * denotes that the best result are statistically significant compared with the best baseline with p value less than 0.95 using matched-pair t-test.

| | | noise $= 20\%$ | | noise $= 40\%$ | | noise $= 60\%$ | | noise $= 80\%$ | |
| | | accuracy | %SAM | accuracy | %SAM | accuracy | %SAM | accuracy | %SAM |
|---|---|---|---|---|---|---|---|---|---|
| *ResNet-18* | ERM | $66.83 \pm 0.21$ | 0.0 | $54.58 \pm 0.96$ | 0.0 | $47.98 \pm 0.36$ | 0.0 | $26.21 \pm 3.40$ | 0.0 |
| | SAM (Foret et al., 2021) | $69.60 \pm 0.19$ | 100.0 | $59.85 \pm 0.53$ | 100.0 | $52.50 \pm 0.25$ | 100.0 | $23.79 \pm 3.21$ | 100.0 |
| | ESAM (Du et al., 2022a) | $75.33 \pm 0.19$ | 100.0 | $67.75 \pm 0.83$ | 100.0 | $4.79 \pm 3.58$ | 100.0 | $1.29 \pm 0.10$ | 100.0 |
| | ASAM (Kwon et al., 2021) | $67.76 \pm 0.86$ | 100.0 | $57.13 \pm 0.06$ | 100.0 | $48.69 \pm 0.04$ | 100.0 | $29.46 \pm 0.10$ | 100.0 |
| | OG-SAM | $75.45 \pm 0.27$ | 100.0 | $68.01 \pm 0.19$ | 100.0 | $56.63 \pm 0.10$ | 100.0 | **29.77**$^* \pm 1.08$ | 100.0 |
| | SS-SAM (Zhao, 2022) | $75.68 \pm 0.62$ | 60.0 | $64.72 \pm 0.20$ | 60.0 | $55.55 \pm 1.49$ | 60.0 | $23.90 \pm 5.63$ | 60.0 |
| | AE-SAM (Jiang et al., 2023) | $68.69 \pm 0.35$ | 61.4 | $57.35 \pm 0.24$ | 61.4 | $47.95 \pm 1.01$ | 61.4 | $27.11 \pm 0.57$ | 61.4 |
| | AO-SAM | **75.69**$^* \pm 0.35$ | 61.2 | **68.35**$^* \pm 0.21$ | 61.3 | **56.95**$^* \pm 1.00$ | 61.2 | $29.76 \pm 1.21$ | 61.3 |
| *ResNet-32* | ERM | $69.33 \pm 0.24$ | 0.0 | $55.77 \pm 0.74$ | 0.0 | $46.96 \pm 0.93$ | 0.0 | $25.67 \pm 0.98$ | 0.0 |
| | SAM (Foret et al., 2021) | $70.88 \pm 0.32$ | 100.0 | $60.40 \pm 0.07$ | 100.0 | $53.10 \pm 0.36$ | 100.0 | $10.66 \pm 5.56$ | 100.0 |
| | ESAM (Du et al., 2022a) | $77.09 \pm 0.22$ | 100.0 | $66.17 \pm 1.78$ | 100.0 | $3.02 \pm 0.26$ | 100.0 | $1.85 \pm 0.73$ | 100.0 |
| | ASAM (Kwon et al., 2021) | $69.64 \pm 1.36$ | 100.0 | $57.88 \pm 0.61$ | 100.0 | $48.79 \pm 0.24$ | 100.0 | $28.06 \pm 1.05$ | 100.0 |
| | OG-SAM | $78.05 \pm 0.23$ | 100.0 | $66.74 \pm 0.19$ | 100.0 | **56.06**$^* \pm 0.13$ | 100.0 | $29.55 \pm 2.08$ | 100.0 |
| | SS-SAM (Zhao, 2022) | $71.34 \pm 0.32$ | 60.0 | $61.45 \pm 1.36$ | 60.0 | $51.76 \pm 0.04$ | 60.0 | $13.96 \pm 3.17$ | 60.0 |
| | AE-SAM (Jiang et al., 2023) | $68.94 \pm 0.12$ | 61.3 | $58.41 \pm 1.89$ | 61.3 | $51.48 \pm 1.08$ | 61.2 | $28.44 \pm 0.76$ | 61.3 |
| | AO-SAM | **78.11**$^* \pm 0.14$ | 61.2 | **68.71**$^* \pm 0.46$ | 61.2 | $54.58 \pm 0.30$ | 61.2 | **29.78**$^* \pm 0.42$ | 61.2 |
| *WideResNet-28-10* | ERM | $74.31 \pm 0.61$ | 0.0 | $62.31 \pm 0.41$ | 0.0 | $48.23 \pm 0.92$ | 0.0 | $29.96 \pm 0.21$ | 0.0 |
| | SAM (Foret et al., 2021) | $76.04 \pm 0.38$ | 100.0 | $64.65 \pm 0.79$ | 100.0 | $56.03 \pm 0.76$ | 100.0 | $29.48 \pm 0.23$ | 100.0 |
| | ESAM (Du et al., 2022a) | $80.06 \pm 0.12$ | 100.0 | $72.03 \pm 0.79$ | 100.0 | $9.75 \pm 2.12$ | 100.0 | $1.16 \pm 0.08$ | 100.0 |
| | ASAM (Kwon et al., 2021) | $74.37 \pm 0.18$ | 100.0 | $62.91 \pm 0.71$ | 100.0 | $51.35 \pm 0.31$ | 100.0 | $33.12 \pm 0.16$ | 100.0 |
| | OG-SAM | $80.14 \pm 0.29$ | 100.0 | **72.79** $^* \pm 0.51$ | 100.0 | **57.01** $^* \pm 0.34$ | 100.0 | **36.33** $^* \pm 1.68$ | 100.0 |
| | SS-SAM (Zhao, 2022) | $75.48 \pm 0.26$ | 60.0 | $64.72 \pm 0.25$ | 60.0 | $54.83 \pm 0.48$ | 60.0 | $35.88 \pm 3.23$ | 60.0 |
| | AE-SAM (Jiang et al., 2023) | $75.46 \pm 0.36$ | 61.3 | $63.04 \pm 0.68$ | 61.3 | $52.29 \pm 0.63$ | 61.3 | $33.72 \pm 0.62$ | 61.3 |
| | AO-SAM | **80.32**$^* \pm 0.07$ | 61.2 | $72.10 \pm 0.48$ | 61.2 | $56.89 \pm 0.30$ | 61.2 | $36.03 \pm 0.59$ | 61.2 |

Table 7: Means and standard deviations of testing accuracy and fraction of SAM updates (%SAM) on *CIFAR-10* and *CIFAR-100*. Results of ERM, SAM, and ESAM are from (Jiang et al., 2023). † denotes that the baseline results are obtained with the authors' provided code. Methods are grouped based on %SAM. The highest accuracy for each network architecture is in bold.

| | | CIFAR-10 | | CIFAR-100 | |
|---|---|---|---|---|---|
| | | Accuracy | % SAM | Accuracy | % SAM |
| *ResNet-18* | ERM | $95.41_{\pm 0.03}$ | $0.0_{\pm 0.0}$ | $78.17_{\pm 0.05}$ | $0.0_{\pm 0.0}$ |
| | SAM (Foret et al., 2021) | $96.52_{\pm 0.12}$ | $100.0_{\pm 0.0}$ | $80.17_{\pm 0.15}$ | $100.0_{\pm 0.0}$ |
| | ESAM (Du et al., 2022a) | $96.56_{\pm 0.08}$ | $100.0_{\pm 0.0}$ | $80.41_{\pm 0.10}$ | $100.0_{\pm 0.0}$ |
| | ASAM (Kwon et al., 2021) | $96.55^{\dagger}_{\pm 0.14}$ | $100.0_{\pm 0.0}$ | $80.52^{\dagger}_{\pm 0.13}$ | $100.0_{\pm 0.0}$ |
| | OG-SAM | $96.79_{\pm 0.02}$ | $100.0_{\pm 0.0}$ | $\mathbf{80.76}_{\pm 0.15}$ | $100.0_{\pm 0.0}$ |
| | SS-SAM (Zhao, 2022) | $96.64^{\dagger}_{\pm 0.02}$ | $60.0_{\pm 0.0}$ | $80.49^{\dagger}_{\pm 0.10}$ | $60.0_{\pm 0.0}$ |
| | AE-SAM (Jiang et al., 2023) | $96.66^{\dagger}_{\pm 0.02}$ | $61.3_{\pm 0.1}$ | $79.96^{\dagger}_{\pm 0.08}$ | $61.3_{\pm 0.0}$ |
| | AO-SAM | $\mathbf{96.82}_{\pm 0.04}$ | $61.1_{\pm 0.0}$ | $80.70_{\pm 0.14}$ | $61.2_{\pm 0.0}$ |
| *WideResNet-28-10* | ERM | $96.34_{\pm 0.12}$ | $0.0_{\pm 0.0}$ | $81.56_{\pm 0.14}$ | $0.0_{\pm 0.0}$ |
| | SAM (Foret et al., 2021) | $97.27_{\pm 0.11}$ | $100.0_{\pm 0.0}$ | $83.42_{\pm 0.05}$ | $100.0_{\pm 0.0}$ |
| | ESAM (Du et al., 2022a) | $97.29_{\pm 0.11}$ | $100.0_{\pm 0.0}$ | $84.51_{\pm 0.02}$ | $100.0_{\pm 0.0}$ |
| | ASAM (Kwon et al., 2021) | $97.38^{\dagger}_{\pm 0.09}$ | $100.0_{\pm 0.0}$ | $84.48^{\dagger}_{\pm 0.10}$ | $100.0_{\pm 0.0}$ |
| | OG-SAM | $\mathbf{97.56}_{\pm 0.03}$ | $100.0_{\pm 0.0}$ | $84.74_{\pm 0.02}$ | $100.0_{\pm 0.0}$ |
| | SS-SAM (Zhao, 2022) | $97.32^{\dagger}_{\pm 0.03}$ | $60.0_{\pm 0.0}$ | $84.39^{\dagger}_{\pm 0.04}$ | $60.0_{\pm 0.0}$ |
| | AE-SAM (Jiang et al., 2023) | $97.37^{\dagger}_{\pm 0.08}$ | $61.3_{\pm 0.0}$ | $84.23^{\dagger}_{\pm 0.08}$ | $61.3_{\pm 0.0}$ |
| | AO-SAM | $97.49_{\pm 0.02}$ | $61.2_{\pm 0.0}$ | $\mathbf{84.80}_{\pm 0.11}$ | $61.2_{\pm 0.0}$ |
| *PyramidNet-110* | ERM | $96.62_{\pm 0.10}$ | $0.0_{\pm 0.0}$ | $81.89_{\pm 0.15}$ | $0.0_{\pm 0.0}$ |
| | SAM (Foret et al., 2021) | $97.30_{\pm 0.10}$ | $100.0_{\pm 0.0}$ | $84.46_{\pm 0.05}$ | $100.0_{\pm 0.0}$ |
| | ESAM (Du et al., 2022a) | $97.81_{\pm 0.01}$ | $100.0_{\pm 0.0}$ | $85.56_{\pm 0.05}$ | $100.0_{\pm 0.0}$ |
| | ASAM (Kwon et al., 2021) | $97.71^{\dagger}_{\pm 0.09}$ | $100.0_{\pm 0.0}$ | $85.55^{\dagger}_{\pm 0.11}$ | $100.0_{\pm 0.0}$ |
| | OG-SAM | $97.79_{\pm 0.04}$ | $100.0_{\pm 0.0}$ | $\mathbf{85.74}_{\pm 0.14}$ | $100.0_{\pm 0.0}$ |
| | SS-SAM (Zhao, 2022) | $97.62^{\dagger}_{\pm 0.03}$ | $60.0_{\pm 0.0}$ | $85.41^{\dagger}_{\pm 0.11}$ | $60.0_{\pm 0.0}$ |
| | AE-SAM (Jiang et al., 2023) | $97.52^{\dagger}_{\pm 0.07}$ | $61.4_{\pm 0.1}$ | $85.43^{\dagger}_{\pm 0.08}$ | $61.4_{\pm 0.1}$ |
| | AO-SAM | $\mathbf{97.87}_{\pm 0.02}$ | $61.2_{\pm 0.0}$ | $85.60_{\pm 0.07}$ | $61.2_{\pm 0.12}$ |
| *ViT-S16* | ERM | $86.69_{\pm 0.11}$ | $0.0_{\pm 0.0}$ | $62.42_{\pm 0.22}$ | $0.0_{\pm 0.0}$ |
| | SAM (Foret et al., 2021) | $87.37_{\pm 0.09}$ | $100.0_{\pm 0.0}$ | $63.23_{\pm 0.25}$ | $100.0_{\pm 0.0}$ |
| | ESAM (Du et al., 2022a) | $84.27_{\pm 0.11}$ | $100.0_{\pm 0.0}$ | $62.11_{\pm 0.15}$ | $100.0_{\pm 0.0}$ |
| | AO-SAM | $\mathbf{88.27}_{\pm 0.12}$ | $100.0_{\pm 0.0}$ | $\mathbf{64.45}_{\pm 0.23}$ | $100.0_{\pm 0.0}$ |

| Method | CIFAR-10 | CIFAR-100 | ImageNet |
|---|---|---|---|
| SAM | 0.05 | 0.1 | 0.05 |
| ESAM | 0.05 | 0.05 | 0.05 |
| ASAM | 0.05 | 0.1 | 1.0 |
| AE-SAM | 0.05 | 0.05 | 0.05 |
| SS-SAM | 0.1 | 0.05 | 0.05 |

Table 8: Values of $\rho$ for Different Methods in Experiments

| | CIFAR-10 with ResNet-18 |
|---|---|
| SAM (Foret et al., 2021) | $96.52 \pm 0.12$ |
| EG-SAM increasing $\rho$ | $94.67 \pm 0.03$ |
| EG-SAM | $96.86 \pm 0.01$ |
| EG-SAM-increasing $\rho$ | $96.79 \pm 0.06$ |

Table 9: Increasing $\rho$ results in CIFAR-10 with ResNet-18.

|  | CIFAR-10 | CIFAR-100 |
|---|---|---|
| GSAM (Zhuang et al., 2022) | $97.42 \pm 0.03$ | $84.48 \pm 0.06$ |
| MSAM (Behdin et al., 2023) | $96.95 \pm 0.04$ | $84.07 \pm 0.06$ |
| OG-SAM | $97.49 \pm 0.02$ | $84.80 \pm 0.11$ |

Table 10: More baselines.

| Method | SAM | EG-SAM | EG+SAM | OG-SAM | OGDA+SAM | SAM+AMO | AO-SAM |
|---|---|---|---|---|---|---|---|
| Time per epoch (Sec.) | 33.90 | 54.73 | 71.24 | 34.97 | 41.73 | 70.06 | 27.52 |

Table 11: Time per epoch.

