# OpenReview forum: "Lookahead Sharpness-Aware Minimization"
_ICLR.cc/2024/Conference — Submitted to ICLR 2024_

### Official Review · Reviewer_1P81 · 2023-10-24

**Soundness:** 3 good
**Presentation:** 3 good
**Contribution:** 3 good
**Rating:** 5
**Confidence:** 4

**Summary:**

-	The paper proposes optimization methods using an extra gradient to improve the performance of SAM. In section 3, the authors propose a maximization with respect to $\varepsilon$, while SAM only focuses on maximization with respect to $w$. Then, the authors propose simple optimization schemes, such as EG-SAM and OG-SAM, to mitigate the computational burden of extra gradient methods.

**Strengths:**

-	The paper has a solid mathematical background to derive the optimization methods. Furthermore, the proposed method can alleviate the negative effects of SAM, which can get stuck at saddle points.
-	The paper combines the proposed method with other SAM-related papers, which can reduce the computational burden of EG-SAM.
-	The proposed method has the same convergence rate as SAM, which does not require additional training epochs.
-	The paper demonstrates the performance improvement of using an additional gradient in a wide range of datasets, outperforming existing methods. The authors include various architectures and datasets, including the noise settings.

**Weaknesses:**

-	Please refer to questions.

**Questions:**

I will happily raise the score if the authors can address the following questions:

1.	I am curious about the motivation behind the proposed method. I agree that using the extra gradient (where $w$ and $\varepsilon$ correspond to $x$ and $y$ in Section 3) might be beneficial for training. Can you clarify why  EG (and correspondingly OG) are straightforward solutions in terms of SAM for escaping saddle points? Or the purpose of EG is to increase the loss of $L(w+\varepsilon)$ by perturbing $\varepsilon$? The loss maximization does not seem to align with performance gain [1].

2.	To push further the question 1, I am not convinced that SAM+EG and EG-SAM (also OG) are not highly correlated because $\varepsilon$ in EG-SAM is not calculated to enlarge the loss with respect to $\varepsilon$. Rather, even though I did not calculate the ODE accurately, the behavior of EG-SAM seems somewhat related to implicit methods in ODEs (thus stable) rather than using explicit methods by using $\hat{w_t}$ obtained from the next step in SGD terms.

3.	I agree that EG-SAM can optimize better in quadratic loss, as shown in Section 4 with ODE. However, the optimization itself is discrete, and some behaviors (such as SGD or catapult effects) cannot be explained in ODE settings. Thus, could the authors demonstrate the effects of EG-SAM in real situations with **SGD** to escape saddle points, such as in Fig. 7 in [2]?

4.	The authors provide information about the effects on computational burden compared to %SAM. However, as SAM and EG-SAM both require 100%, they actually differ in terms of computational time. Therefore, wall clock computational time is needed to assess the computational overhead during real training.

[1] https://arxiv.org/abs/2206.06232

[2] https://arxiv.org/abs/2301.06308

**Details Of Ethics Concerns:**

No ethical concerns.

---

> ### Author Response · Authors · 2023-11-20
> **Responses to Reviewer 1P81 (1/2)**
>
> Thank you for your constructive reviews. We summarize the reviewer's questions and present our responses below.
>
>
> ---
> ***Q1: I am curious about the motivation behind the proposed method. ... i) Can you clarify why EG (and correspondingly OG) are straightforward solutions in terms of SAM for escaping saddle points?
> ii) Or the purpose of EG is to increase the loss of $L(w+\varepsilon)$ by perturbing $\varepsilon$ ? The loss maximization does not seem to align with performance gain [1].***
>
>
> i) The lookahead mechanism is designed to help to avoid saddle point by gathering more information about the landscape, as mentioned in Sec. 1. Additionally, another more intuitive explanation is that, as also pointed out by Reviewer hY3b, the EG-SAM update scheme can be written as
> $w_t =w_{t-1}-\eta_t \nabla L( w_{t-1}+\epsilon^\prime_t)$ where $\epsilon^\prime_t = (\rho /||\nabla L(w_{t-1})||-\eta_t) \nabla L(w_{t-1})$.
> Note that SAM's update scheme is
> $w_t =w_{t-1}-\eta_t \nabla L( w_{t-1}+\epsilon_t)$, where
> $\epsilon_t = (\rho/||\nabla L(w_{t-1})||)\nabla L(w_{t-1})$.
> EG-SAM can be interpreted as suppressing perturbation in the early stages and allowing more perturbation in the final stages (as $\epsilon_t$ decreases from large to small values). Since a larger perturbation can make the model more prone to being trapped in saddle points, as indicated by Kim et al. (2023), our method is more effective in avoiding saddle points than SAM.
>
> ii)  Our EG does not aim to increase the loss of $L(w+\varepsilon)$. As discussed in 1), our work is designed to avoid saddle points, which is  different from aiming to increase the loss value of $L(w+\varepsilon)$.
>
> ---
>
> ***Q2: To push further the question 1, I am not convinced that SAM+EG and EG-SAM (also OG) are not highly correlated because $\epsilon$
>  in EG-SAM is not calculated to enlarge the loss with respect to $\epsilon$.***
>
> We agree that SAM+EG can enlarge the losses, as $\epsilon$ is no longer a one-step gradient ascent. However, as discussed in A1, different from SAM+EG, EG-SAM is not designed for enlarging the loss. Thus, SAM+EG and EG-SAM are different.
>
> Specifically, recall SAM can be formulated as a minimax problem (Eq. (1)) $\min_{\boldsymbol{w}} \max_{\boldsymbol{\epsilon}:\|\boldsymbol{\epsilon}\| \leq \rho} L(\boldsymbol{w}+\boldsymbol{\epsilon})$.
> As EG is used for addressing minimax problem,
> SAM+EG can be directly implemented as the solver. However, for EG-SAM, it firstly uses a max oracle
> $  \max_{\boldsymbol{\epsilon}:\|\boldsymbol{\epsilon}\| \leq \rho} L(\boldsymbol{w}+\boldsymbol{\epsilon}) \approx L(\boldsymbol{w}+\frac{\rho \nabla_{\boldsymbol{w}} L(\boldsymbol{w})}{\|\nabla_{\boldsymbol{w}} L(\boldsymbol{w})\|}) $
> following the original SAM (Foret et al., 2020), then (Eq. 1) is converted to a minimization problem $\min_{\boldsymbol{w}} L(\boldsymbol{w}+\frac{\rho \nabla_{\boldsymbol{w}} L(\boldsymbol{w})}{\|\nabla_{\boldsymbol{w}} L(\boldsymbol{w})\|}) $, and finally EG is designed to solve this problem.
>
> ---
> ***Q3: Rather, even though I did not calculate the ODE accurately, the behavior of EG-SAM seems somewhat related to implicit methods in ODEs (thus stable) rather than using explicit methods by using $\hat{w}_t$
>  obtained from the next step in SGD terms.***
>
> Regarding the ODE, we thank the reviewer for sharing the ODE perspective to help explain our method. Indeed, we agree with the reviewer that using future gradients to update the current model is somewhat similar to the implicit methods in ODE. We will further explore this connection in our future works.
>
> ---
> ***Q4: I agree that EG-SAM can optimize better in quadratic loss, as shown in Section 4 with ODE. However, the optimization itself is discrete, and some behaviors (such as SGD or catapult effects) cannot be explained in ODE settings. Thus, could the authors demonstrate the effects of EG-SAM in real situations with SGD to escape saddle points, such as in Fig. 7 in [2]?***
>
> The reviewer might have some misunderstandings. We have already conducted a
> toy
> experiment at the beginning of  Sec. 3, which uses quadratic objective to show the discrete time behaviors of SAM and our methods around saddle point following (Compagnoni et al., 2023). Results reveal that discrete time EG-SAM can escape saddle point faster than SAM.
> Fig. 7 in [2], as referred to by the reviewer,  similarly illustrates the discrete time behavior of SAM around saddle point.
>
> Besides, ODE is also widely used to analyse SAM (Kim et al., 2023; Compagnoni et al., 2023), and we follow the same analysis scheme.

---

> > ### Author Response · Authors · 2023-11-20
> > **Responses to Reviewer 1P81 (2/2)**
> >
> > (Continued)
> >
> > ---
> > ***Q5: The authors provide information about the effects on computational burden compared to \%SAM. However, as SAM and EG-SAM both require 100\%, they actually differ in terms of computational time. Therefore, wall clock computational time is needed to assess the computational overhead during real training.***
> >
> >
> > Note that for EG-SAM, the \%SAM is 150\% rather than 100\%.
> > We guess the reviewer is talking about OG-SAM, whose \%SAM is 100\%, the same as SAM.
> > We also provide  clock computational time as suggested by the reviewer. The following table is the clock computational time per epoch on CIFAR-10 with Resnet-18 as the  backbone.
> > As can be seen, the higher \% SAM ratio means more clock computational time, and our proposed OG-SAM has relatively same training time as SAM, while AO-SAM is faster than SAM.
> >
> >
> >
> >
> >
> > | Method   | SAM   | EG-SAM | EG+SAM | OG-SAM | OGDA+SAM | SAM+AMO | AO-SAM |
> > |----------|-------|--------|--------|--------|----------|---------|--------|
> > | Time per epoch (Sec.) | 33.90 | 54.73 | 71.24 | 34.97 | 41.73 | 70.06 | 27.52 |

---

> > > ### Comment · Reviewer_1P81 · 2023-11-22
> > > **Rebuttal follow-up**
> > >
> > > I have carefully reviewed the authors' rebuttal and the feedback from other reviewers. I thank for the authors for their comprehensive response and the additional effort they have put into enhancing the manuscript. Specifically, I correct my misunderstanding about Q1 and Q2. Still, I argue that some points should be further investigated as reviewer hY3b pointed out. Therefore, I will maintain my current score for the manuscript and await the input of other reviewers.

---

> > > > ### Author Response · Authors · 2023-11-22
> > > > **Thank you for your follow-up comments**
> > > >
> > > > Thank you for your follow-up comments. We are glad to hear that our responses have clarified some of your concerns.
> > > >
> > > > Regarding the points highlighted by Reviewer hY3b, we have added follow-up responses to Reviewer hY3b. We hope these follow-up responses could also address your concerns. Thank you once again for your constructive feedback.

---

### Official Review · Reviewer_gBpE · 2023-10-25

**Soundness:** 3 good
**Presentation:** 2 fair
**Contribution:** 2 fair
**Rating:** 5
**Confidence:** 4

**Summary:**

This paper proposes a new version of multi-step SAM. They discuss convergence properties of the algorithm and present numerical experiments.

**Strengths:**

The proposed method seems to outperform SAM in the settings studied.

**Weaknesses:**

- The idea of multi-step SAM is not new and has been explored before. The EG and OG versions proposed here are new, but I think the contribution of the work is marginal.

- The theory is nice to have, but is not really insightful and the proofs are fairly standard.

- I'm not fully convinced with the numerical experiments. They consider CIFAR + ResNet, and only one example of ImageNet + ResNet. No experiments on transformers or language models are done, which makes the experiments rather limited. Also, important version of SAM such as gSAM (https://arxiv.org/pdf/2203.08065.pdf) and mSAM (https://arxiv.org/pdf/2302.09693.pdf) are not included in the experiments. Also, I didn't find runtimes of methods, which I might've missed.

- The presentation can improve. There are so many versions and abbreviations (EG,OG, AO, AMO) that it's difficult to see the points the authors are trying to make.

**Questions:**

N/A

---

> ### Author Response · Authors · 2023-11-20
> **Responses to Reviewer gBpE**
>
> Thank you for your all valuable suggestions. Below we respond to your key concerns point by point:
>
> ---
> ***Q1: The idea of multi-step SAM is not new and has been explored before. The EG and
> OG versions proposed here are new, but I think the contribution of the work is
> marginal.***
>
> i) The reviewer might have some misunderstandings. Our EG-SAM significantly diverges from that of multi-step SAM methods like those in [1]:
>
> [1] aims to use multi-step SAM to yield a more precise solution of
> max oracle $\boldsymbol{\epsilon}^* (\boldsymbol{w}):= \arg \max_{\boldsymbol{\epsilon}:\|\boldsymbol{\epsilon}\| \leq \rho} L(\boldsymbol{w}+\boldsymbol{\epsilon})$.  This method focuses on refining the maximization step within the SAM framework.
>
> In contrast, our EG-SAM is primarily focused on avoiding saddle points and finding a better trajectory for convergence. This is achieved by incorporating a look-ahead mechanism. Unlike multi-step SAM (e.g., [1]), it is not designed specifically to refine the maximization step but rather to enhance overall convergence by avoiding saddle points.
>
> ii) Regarding the contributions,
> 1) we are the first to improve SAM by using look-ahead mechanism to avoid saddle points.
>
> 2) We design different variants of look-ahead mechanism, e.g. EG-SAM, OG-SAM and AO-SAM, to help further reduce computational cost.
>
> 3) We provide theoretical guarantees.
>
> 4) Our method outperforms other SOTA baselines consistently with different structures and in different datasets.
>
> ---
> ***Q2: The theory is nice to have, but is not really insightful and the proofs are fairly standard.***
>
> i) Regarding our proof is not really insightful,  we provide two distinct properties of EG-SAM: 1) EG-SAM can escape saddle points
> faster than original SAM (Proposition 4.1), and 2) EG-SAM has more chance to converge at
> flatter minima (Proposition 4.2). These findings are novel and were not present in any previous work related to SAM. The ability to escape saddle points quickly and the increased probability of converging to flatter minima provide valuable insights into the advantages of our proposed methods.
>
> ii) Regarding our proof is fairly standard, note that original SAM proof cannot be directly used to prove our proposed methods due to the following reasons:
>
> 1) Recall in  OG-SAM, we have $\hat{w_t}=w_{t-1}-\eta_t g_{t-1}$, which includes past gradient $g_{t-1}$. This makes the standard SAM analysis (Andriushchenko \& Flammarion, 2022) difficult to apply directly, as their analytical approach does not include a technique to address past gradients.
>
> 2) In the proof of AO-SAM, we use an adaptive policy with OG (Jiang et al., 2023). Although (Jiang et al., 2023) provides a proof, it is only established for unnormalized SAM. Since ours is normalized SAM with OG, it inherits the difficulties of both the adaptive policy in normalized SAM and past gradient analysis.
>
> ---
> ***Q3: I'm not fully convinced with the numerical experiments. They consider CIFAR + ResNet, and only one example of ImageNet + ResNet. No experiments on transformers or language models are done, which makes the experiments rather limited. Also, important version of SAM such as gSAM  and mSAM are not included in the experiments. Also, I didn't find runtimes of methods, which I might've missed.***
>
> 1) We added an experiment using vision transformer ViT-S16 on CIFAR-10 and CIFAR-100 in our response to Reviewer rtXV Q3. As can been seen, our proposed method also outperforms the baselines.
>
> 2) We also added the performance comparison between GSAM [3], MSAM [4], and our AO-SAM
> in CIFAR-10 and CIFAR-100 with WideResNet-28-10 backbone, as suggested by the
> reviewer. From the below table, we find that
> our method also outperforms GSAM and MSAM.
>
> |     | CIFAR-10      | CIFAR-100   |
> |---------|-----------------|-----------------|
> | GSAM [3]| 97.42 ± 0.03  | 84.48 ± 0.06    |
> | MSAM [4]| 96.95 ± 0.04  | 84.07 ± 0.06   |
> | OG-SAM  | 97.49 ± 0.02  | 84.80 ± 0.11   |
>
> 3)  We have also added the running time comparisons in our response to Reviewer 1P81's Q5. As can be seen, our proposed OG-SAM
> has relatively same training time as SAM, while AO-SAM is faster than SAM.
>
> ---
> ***Q4: The presentation can improve. There are so many versions and abbreviations (EG, OG, AO, AMO) that it's difficult to see the points the authors are trying to make.***
>
> Thank you for pointing out. we have included the full names alongside their abbreviations wherever necessary to enhance readability.
>
> ---
> Reference:
>
> [1] Kim, Hoki, et al. "Exploring the effect of multi-step ascent in sharpness-aware minimization." arXiv preprint arXiv:2302.10181 (2023).
>
> [2] Dosovitskiy, Alexey, et al. "An Image is Worth 16x16 Words: Transformers for Image Recognition at Scale." ICLR. 2020.
>
> [3] Zhuang, Juntang, et al. "Surrogate Gap Minimization Improves Sharpness-Aware Training." ICLR. 2021.
>
> [4] Behdin, Kayhan, et al. "mSAM: Micro-Batch-Averaged Sharpness-Aware Minimization." arXiv preprint arXiv:2302.09693 (2023).

---

> > ### Comment · Reviewer_gBpE · 2023-11-22
> >
> > Thank you for your response.
> >
> > - The mentioned propositions are based on two strong assumptions: infinitesimal step sizes (ODE approximation) and quadratic loss, neither of which holds true in practice. Therefore, I don't think they theory can support the claims made.
> >
> > - I think the new experimental results look interesting, with the addition of these results to the paper, I increase my score.

---

> > > ### Author Response · Authors · 2023-11-23
> > > **Thank you for your follow-up comments**
> > >
> > > Thank you for increasing score! We appreciate your follow-up comments and would like to respond as follows:
> > >
> > >
> > >
> > >
> > > ---
> > >
> > > ***Q5: The propositions mentioned are based on two strong assumptions: infinitesimal step sizes (ODE approximation) and quadratic loss, neither of which are typically present in practical scenarios. Therefore, I am skeptical about the ability of this theory to support the claims made.***
> > >
> > >
> > > The assumptions of infinitesimal step sizes and quadratic loss directly follow the work in [1], which is also used in the analysis of SAM's behavior.
> > >
> > > In addition to the theoretical analysis, we conducted an experiment detailed at the beginning of Section 3. This experiment demonstrates that our EG-SAM method escapes saddle points more rapidly than the standard SAM, providing practical evidence to support our theoretical analysis.
> > >
> > > ---
> > >
> > > ***Reference:***
> > >
> > > [1] Compagnoni, Enea Monzio, et al. "An SDE for Modeling SAM: Theory and Insights." ICML 2023.

---

### Official Review · Reviewer_hY3b · 2023-10-26

**Soundness:** 3 good
**Presentation:** 2 fair
**Contribution:** 1 poor
**Rating:** 3
**Confidence:** 4

**Summary:**

- The paper argues that sharpness-aware minimization can be stuck in saddle-points (of the loss function).
- This motivates the EG-SAM scheme which modifies SAM by perturbing an *extrapolated* point (computed with the same gradient information as the perturbation).
- Their alternative (optimistic) variant, OG-SAM, replaces the gradient in the extrapolation update with the gradient at the previous perturbed point.
- They combine both methods with an existing adaptive policy (AE-SAM) and show favorable numerics on CIFAR10, CIFAR100 and ImageNet.
- Theoretically, they provide a convergence result under gradient-Lipschitz and bounded variance for the average gradient norm with decreasing stepsize and perturbation radius $\rho$.
- They also show that an ODE related to EG-SAM has a smaller region of attraction then the SAM ODE.

**Strengths:**

- The paper is easy to follow
- It provides a good literature overview

**Weaknesses:**

I have a range of concerns starting with the formulation itself:

- The EG-SAM update can be rewritten as:

    $\tilde w_t = w_{t-1} + (\rho/\|\nabla L(w_{t-1})\| - \eta_t) \nabla L(w_{t-1})$

    $\tilde w_t = w_{t-1} - \eta_t \nabla L(\tilde w_{t})$

    So when the stepsize is taken decreasing (as in the experiments) it is simply a scheme that perturbs less adversarially in the early training phase and more later. It would be instructive to plot the gradient norm at $w_{t-1}$ throughout training. How does EG-SAM compare against SAM with an (increasing) stepsize schedule for $\rho$?
- OG-SAM appears unnecessary since it does not save on gradient computations: EG-SAM simultaneous computes the extrapolation and perturbation computation so it already only requires two gradient computations. What is the motivation for OG-SAM in not saving gradient computations?

Theoretically:

- The convergence result in Section 4.2 requires strong assumptions on both the problem class and the a decreasing perturbation radius $\rho$. Requiring both gradient-Lipschitz and function Lipschitz simultaneously is quite restrictive. More importantly, by decreasing $\rho$ you essentially avoid the difficulty of the perturbation. One indication that your assumptions are too strong is that your argument would hold for any kind of (feasible) perturbation.

- I appreciate the attempt to contextualize the convergence result in Section 4.1 within the literature for extragradient (EG), but I find the comparisons quite misleading. Stating that their assumptions are restrictive whereas your assumption are more general is problematic. They are different problems entirely for multiple reasons:
    - Different solution concepts: The EG literature treats convergence to a equilibrium, whereas the EG-SAM results only treats the gradient of the minimizing player.
    - Different application: You apply EG to the minimization problem (not to treat the minimax)
    The same issue appears in other sections when comparing rates:
    - Section 3.1: even if assuming the loss in Eq. 1 is convex in $w$ then it is also *convex* in $\varepsilon$, so the rate of $\mathcal O(T^{-1/4})$ does not apply (we could still hope to be locally linear maybe for the max problem). The rates for SAM that are being compared against is for the gradient norm (so it is not considering convergence of the max-player).
    - Section 3.2 mentions that the SAM+OG rate is much slower than SAM, but you are comparing both different problem settings (nonconvex-strongly concave vs nonconvex-nonconvex) and performance metrics.

- The analysis in section 4.1 seems to follow directly from Compagnoni et al. 2023 Lemma 4.3: Eq. 19 is simply the SAM ODE (Eq. 18) with a different stepsize. More importantly, the extrapolation stepsize $\eta$ in Eq. 19 should arguably not be fixed. Otherwise it corresponds to a discrete scheme which takes the extrapolation stepsize much larger than the actual stepsize.

Empirically:

- I appreciate the effort of providing standard deviations for the experiments, but it is arguably even more important to sweep over $\rho$ when comparing different SAM variants. The optimal $\rho$ can differ significantly across models/datasets and can influence the ordering of the methods significantly.


Minor:

- $\nabla F$ notation can be misleading since it is not a gradient. I suggest simply letting $F(z)=(\nabla_x f(x,y), -\nabla_y f(x,y))$
- Maybe explicitly mention that the number of gradient oracles are 4 in Eq. 12 (instead of saying "two approximated gradient steps")
- line 4 in Algorithm 3: specify what the AE-SAM update is

Typos:

- Eq. 9 and 10 has a missing stepsize for $\varepsilon$
- Second equation of section 3.2 has a missing hat and stepsize
- Theorem 4.6 misses an index $t$ for $\rho$

**Questions:**

- What is the value of $\rho$ used for the different methods in the experiments?
- Why is EG-SAM 150% SAM in Table 1?
- Can you plot $\|\nabla L(w_{t-1})\|$ throughout training? (or a mini-batch estimate)
- Section 3.3 what is meant by "has an approx solution [...] so one can directly apply extra-gradient". Are you arguing based on Danskin's theorem? (the cited reference doesn't seem to mention the same problem)

---

> ### Author Response · Authors · 2023-11-20
> **Responses to Reviewer hY3b (1/3)**
>
> We thank the reviewer’s helpful feedback on our submission. We summarize the reviewer’s questions and the following is our responses:
>
> ---
>
> ***Q1:
> The EG-SAM update can be rewritten as:
> ... When the stepsize is taken decreasing (as in the experiments) it is simply a scheme that perturbs less adversarially in the early training phase and more later.***
>
>
> Our scheme is designed based on some non-trivial insights:
>
> i) This scheme is based on incorporating the lookahead mechanism into SAM, which has not been proposed previously.
>
> ii) The scheme is also explainable. By suppressing perturbation during the early stages, our method is more effective at avoiding saddle points than the original SAM. This is because larger perturbations make the model more susceptible to being trapped in saddle points, as noted by Kim et al. (2023). Additionally, by allowing more perturbation in the final stages, our method encourages the model to find flat minima.
>
> ---
>
> ***Q2: It would be instructive to plot the gradient norm at $w_t$
>  throughout training.***
>
> We added gradient norm at $\nabla L(w_t)$ in Figure 5, Sec. B, as suggested by the reviewer. As can be seen by the figure, the gradient of EG-SAM is larger than that of SAM in the early stages, but it converges to the same level as SAM in the final stage.
>
> ---
>
> ***Q3: How does EG-SAM compare against SAM with an (increasing) stepsize schedule for $\rho$?***
>
> We added an extra experiment regarding using increasing stepsize schedule for $\rho$, i.e., $\rho_t = \rho_0+  \frac{t(\rho_{T}- \rho_0)}{T}$, where $\rho_{T}$ is the final value
> $\rho_{T}$ (1.0 in our experiment), while $\rho_0$ is the initial value of $\rho$ (0.1 in our experiment).
> As can been from the table, our method with increasing $\rho$ still outperforms the SAM, and is comparable to original EG-SAM, while SAM with increasing $\rho$ perform worse than original SAM method.
>
>
>
> |                              | CIFAR-10 with ResNet-18     |
> |------------------------------|-----------------------------|
> | SAM (Foret et al., 2021)     | 96.52 ± 0.12                |
> | SAM increasing $\rho$     | 94.67 ± 0.03                |
> | EG-SAM                       | 96.86 ± 0.01                |
> | EG-SAM-increasing $\rho$     | 96.79 ± 0.06                |
>
>
> ---
>
> ***Q4: OG-SAM appears unnecessary since it does not save on gradient computations: EG-SAM simultaneous computes the extrapolation and perturbation computation so it already only requires two gradient computations. What is the motivation for OG-SAM in not saving gradient computations?***
>
> In EG-SAM, we need to compute three variables at each step: $\hat{w}$, $\hat{\epsilon}$, and $w$. In contrast, standard SAM and OG-SAM require only $\hat{\epsilon}$ and $w$ per step (OG-SAM reuses the gradient of  $\hat{w}$), and thus we are saving gradients.
> Besides this computational efficiency, another advantage of OG-SAM is that it contains the past gradient
> $\boldsymbol{g}_{t-1}$ at current step $t$. As discussed in Sec. 3.4, by leveraging extra information from the past gradient, it improves the performance (Rakhlin \& Sridharan, 2013).
>
> ---
> ***Q5: The convergence result in Section 4.2 requires strong assumptions on both the i) problem class and ii) the a decreasing perturbation radius
> $\rho$. ...
> More importantly, by decreasing
> $\rho$ you essentially avoid the difficulty of the perturbation. One indication that your assumptions are too strong is that your argument would hold for any kind of (feasible) perturbation.***
>
>
> i) Regarding the problem class assumptions, note that we do not introduce the function Lipschitz assumption. Our three assumptions are bounded variance, smoothness (gradient Lipschitz), and uniformly bounded gradient (Assumptions 4.3-4.5). These assumptions are widely used in the analysis of SAM and its variants (Mi et al., 2022; Dai et al.,
> 2023; Zhang et al., 2023; Yue et al., 2023). Beyond SAM, these assumptions are utilized in analyzing different optimizers [3].
>
>
>
> ii) Regarding decreasing $\rho$, a recent paper [1] theoretically reveals that
> SAM with constant $\rho$ CANNOT converge to stationary points. Therefore, assuming a decreasing
> $\rho$ is necessary in theoretical analysis. Moreover, related works with decreasing $\rho$ assumptions under stochastic gradient settings  are also found in related works (Mi et al., 2022; Dai et al.,
> 2023; Zhang et al., 2023; Yue et al., 2023).

---

> > ### Author Response · Authors · 2023-11-20
> > **Responses to Reviewer hY3b (2/3)**
> >
> > (Continued)
> >
> > ---
> > ***Q6: ... but I find the comparisons to extragradient (EG) quite misleading. Stating that their assumptions are restrictive whereas your assumption are more general is problematic. They are different problems entirely for multiple reasons:***
> >
> > ***i) Different solution concepts.***
> >
> > ***ii)
> > Different application.***
> >
> > ***iii)
> > Section 3.1: even if assuming the loss in Eq. 1 is convex in $w$
> >  then it is also convex in $\epsilon$
> > , so the rate of $\mathcal{O}\left(T^{-1 / 4}\right)$
> >  does not apply. The rates for SAM that are being compared against is for the gradient norm.***
> >
> > ***iv)
> > Section 3.2 mentions that the SAM+OG rate is much slower than SAM, but you are comparing both different problem settings (nonconvex-strongly concave vs nonconvex-nonconvex) and performance metrics.***
> >
> > Before addressing the specific points raised, we should highlight that EG-SAM still aims to find the solution of the SAM's minimax problem (Eq. (1)). To accelerate convergence, EG-SAM utilizes an approximated max oracle, turning the minimax problem into the minimization problem. To validate this,
> > we add a new section (Sec. A.5) to explain the relationship between stationary point found by EG-SAM ($\mathbb{E}|| \nabla L(\boldsymbol{w}_{t})|| ^{2} =0$)
> >
> > and the stationary point for the minimax problem Eq. (1) ($\mathbb{E}||\nabla_{\boldsymbol{w}} \max_{\epsilon,||\epsilon||\leq \rho} L(\boldsymbol{w} +\epsilon) ||^{2}=0$):
> >
> > If $\boldsymbol{w}_t^{*} \in \\{ \boldsymbol{w}_t \| \mathbb{E}|| \nabla L(\boldsymbol{w}_t) ||^{2} =0\\}$,
> >
> > then
> > $\mathbb{E} ||\nabla \max_{\epsilon,||\epsilon ||\leq \rho } L(\boldsymbol{w}_t^{*} +\epsilon) ||^{2} \leq \beta ^{2} \rho ^{2}$, where $\beta$ is the parameter for $\beta$ smoothness.
> >
> > We agree that the converged stationary point found by EG-SAM may not be the stationary point of (Eq. (1)). However, when $\rho$ is not large, the gap is small.  In practical experiments, $\rho$ is typically small (normally less than 0.1), making the distance small.
> > This is further corroborated by empirical evidence (Foret et al., 2020) showing that more precise max oracles, such as multi gradient ascent perturbation steps, do not improve the performance.
> >
> > The following is our point-to-point response:
> >
> > i) As discussed, EG-SAM finds a stationary point whose distance to the stationary point of the original SAM's minimax problem is small.
> >
> > ii) As discussed, EG-SAM finds a stationary point close to the stationary point of original SAM's minimax problem. Since the convergence rate for other EG-based methods is also for the stationary point of a minimax problem (Lin et al., 2020b; Mahdavinia et al., 2022),
> > EG-SAM's convergence rate remains comparable to other EG-based methods.
> >
> > iii)
> > Regarding the reviewer's inquiry about  1) EG can achieve a local linear convergence rate, and 2) our convergence rate is for the gradient norm of the minimization problem rather than the minimax problem:
> >
> >
> > 1) we agree that EG can achieves linear convergence rate for deterministic EG in the convex-concave minimax problem. However, SAM is a non-convex and non-concave problem with a stochastic gradient. To the best of our knowledge, no EG-based methods have been shown to achieve a linear convergence rate under this setting.
> >
> >
> > 2) note that the rate of $\mathcal{O}(T^{-1 / 4})$ is for the
> > stationary point of a minimax problem (Lin et al., 2020b; Mahdavinia et al., 2022).  As discussed, EG-SAM also finds an approximated stationary point for the minimax problem, making the convergence rate of the gradient norm a suitable metric for comparison between EG-SAM and the original EG.
> >
> > iv) (Stochastic) EG and OG's convergence rates are $\mathcal{O}(T^{-1 / 4})$ under the
> > nonconvex strongly-concave setting. Thus,
> > the current convergence rate for directly using EG/OG (EG+SAM and OG+SAM) to solve SAM's minimax problem Eq. (1) is also $\mathcal{O}(T^{-1 / 4})$ under the
> > nonconvex strongly-concave setting. However, for SAM (also our EG-SAM and OG-SAM), our proofs do not rely on extra assumptions regarding the convexity or concavity of Eq. (1). Thus, our framework addresses a nonconvex-nonconcave minimax problem, which is even weaker assumptions than SAM+EG/SAM+OG. As discussed in 3), the gradient norm a suitable metric. Therefore, comparing the convergence rate is still fair.

---

> > > ### Author Response · Authors · 2023-11-20
> > > **Responses to Reviewer hY3b (3/3)**
> > >
> > > (Continued)
> > >
> > > ---
> > > ***Q7:The analysis in section 4.1 seems to follow directly from Compagnoni et al. 2023 Lemma 4.3: Eq. 19 is simply the SAM ODE (Eq. 18) with a different stepsize. More importantly, the extrapolation stepsize $\eta$
> > >  in Eq. 19 should arguably not be fixed. Otherwise it corresponds to a discrete scheme which takes the extrapolation stepsize much larger than the actual stepsize.***
> > >
> > > We thank the reviewer's suggestion regarding the fixed extrapolation stepsize. We agree with the reviewer assuming $\eta$ is fixed is not ideal. Consequently, we have revised the proofs in Section A.1 to accommodate a varying extrapolation stepsize $\eta_\tau$ to vary w.r.t. the time $\tau$. Formally, we now define $\eta_{\tau} : \mathbb{R}^+ \to \mathbb{R}^+$ as a function that satisfies the condition $\eta_{\tau } \geq \eta_{\tau+\epsilon'} > 0$ for every $\tau \in \mathbb{R}^+$ and $\epsilon' \in \mathbb{R}^+$.
> > >
> > > We also carefully re-evaluated our main results (Propositions 4.1 and 4.2) under this new framework and found that our main results (Prop 4.1 and 4.2) remain valid. We have updated the new proofs in the revised version.
> > >
> > > ---
> > > ***Q8: I appreciate the effort of providing standard deviations for the experiments, but it is arguably even more important to sweep over
> > > $\rho$ when comparing different SAM variants. The optimal $\rho$
> > >  can differ significantly across models/datasets and can influence the ordering of the methods significantly.***
> > >
> > > In our experiments, we utilized $\rho$ values as specified in the original papers for each method. These values represent the optimal $\rho$ as determined by sensitivity analyses conducted in those studies.
> > >
> > > ---
> > > ***Q9: What is the value of $\rho$ used for the different methods in the experiments?***
> > >
> > > A9: For the experiments, we adhered to the $\rho$ values mentioned in the original papers for each method. Below is a table summarizing these values across different datasets. We also added it in our revised version.
> > >
> > >  | **Method** | **CIFAR-10** | **CIFAR-100** | **ImageNet** |
> > > |------------|--------------|---------------|--------------|
> > > | SAM        | 0.05         | 0.1           | 0.05         |
> > >  | ESAM       | 0.05         | 0.05          | 0.05         |
> > > | ASAM       | 0.05         | 0.1           | 1.0          |
> > > | AE-SAM     | 0.05         | 0.05          | 0.05         |
> > > | SS-SAM     | 0.1          | 0.05          | 0.05         |
> > >
> > >  ---
> > > ***Q10:Maybe explicitly mention that the number of gradient oracles are 4 in Eq. 12 (instead of saying "two approximated gradient steps")***
> > >
> > > We think Eq. 12 only has two gradient oracles,
> > > as $\hat{\boldsymbol{\epsilon}}_t$ and $\boldsymbol{\epsilon}'_t$ are approximated max oracles and not gradient oracles.
> > >
> > > ---
> > > ***Q11:
> > > $\nabla F$ notation can be misleading; Line 4 in Algorithm 3: specify what the AE-SAM update is; typos: Eq. 9 and 10 has a missing stepsize for $\varepsilon$;
> > > Second equation of section 3.2 has a missing hat and stepsize
> > > Theorem 4.6 misses an index $t$ for $\rho$.***
> > >
> > >
> > > Thank you for pointing out these issues. We have thoroughly reviewed and corrected each of them in the revised version of our paper.
> > >
> > > ---
> > > ***Q12: Why is EG-SAM $150$\% SAM in Table 1?***
> > >
> > > Recall the definition of the SAM ratio %SAM$\equiv 100 \times (\sum^T_{t=1} \text{numbers of SAM used at epoch t})/T$.
> > > For EG-SAM, each step involves 1.5 times the SAM calculations compared to standard SAM. This is because, in EG-SAM, we need to compute three quantities at each step: $\hat{w}$, $\hat{\epsilon}$, and $w_t$. In contrast, standard SAM requires only $\hat{\epsilon}$ and $w_t$ per step. Therefore, the SAM ratio for EG-SAM is 150\%.
> > >
> > > ---
> > > ***Q13: Section 3.3 what is meant by "has an approx solution [...] so one can directly apply extra-gradient". Are you arguing based on Danskin's theorem? (the cited reference doesn't seem to mention the same problem)***
> > >
> > >
> > > The approximation means $  \max_{\boldsymbol{\epsilon}:\|\boldsymbol{\epsilon}\| \leq \rho} L(\boldsymbol{w}+\boldsymbol{\epsilon}) \approx L(\boldsymbol{w}+\frac{\rho \nabla_{\boldsymbol{w}} L(\boldsymbol{w})}{\left\|\nabla_{\boldsymbol{w}} L(\boldsymbol{w})\right\|}) $, which represents the approximated max oracle. This approximation is widely used in SAM and its variants (Foret et al., 2021; Kwon et al., 2021).
> > > By using this approximation, we convert the minimax problem (Eq. (1)) into a minimization problem, as also discussed in our responses to Q6. Then, this minimization problem can be addressed using the extra-gradient method. The citation (Leng et al., 2018) simply illustrates a variant of the extra-gradient method.
> > >
> > > ---
> > > References
> > >
> > > [1] Si, Dongkuk, and Chulhee Yun. "Practical Sharpness-Aware Minimization Cannot Converge All the Way to Optima." NeurIPS 2023.
> > >
> > > [2] Jin, Chi, Praneeth Netrapalli, and Michael Jordan. "What is local optimality in nonconvex-nonconcave minimax optimization?." ICML 2020.
> > >
> > > [3] Défossez, Alexandre, et al. "A Simple Convergence Proof of Adam and Adagrad." TMLR (2022).

---

> > > > ### Comment · Reviewer_hY3b · 2023-11-21
> > > >
> > > > I thank the reviewers for the thorough response. However, I don't find that it addresses my primary concerns, so I choose to maintain my score. In order to highlight what some of those are:
> > > >
> > > >
> > > > - **Q4, Q12** I believe there is a big misunderstand concerning what is the computational bottleneck in SAM. The authors seems to count how many variables they maintain, but what matter computationally is the number of backpropagations. EG-SAM and OG-SAM seems to have the same number of backpropagations per iterations, in which case the motivation of OG-SAM remains unclear to me.
> > > >
> > > > - _**Q3**_ For the (new) increasing $\rho$ baseline, it seem like the choice of $\rho$  is different from the other methods? The intention was to see whether increasing $\rho$ (which can become *identical* to EG-SAM appropriate stepsize schedule) can perform similarly, so you need to at least pick $\rho$ similarly.
> > > >
> > > > - _**Q5**_ You *do* implicitly assume function Lipschitz by assuming bounded variance.
> > > >
> > > > - _**Q6**_ I seems that the authors are misunderstanding my concern. The rates that the authors mentions concerning EG do not apply to their problem (which is *convex* in the max-player and not (strongly)-concave as otherwise assumed) and vice versa. I would be more careful with drawing parallels to classical minimax optimization as this can be misleading.
> > > >
> > > > - _**Q9**_ What is the choice of $\rho$ of *your* method? You seems to search over a much more refined grid (0.01, 0.05, 0.08, 0.1, 0.5, 0.8, 1, 1.5, 1.8, 2), possibly explaining the difference in performance.

---

> > > > > ### Author Response · Authors · 2023-11-22
> > > > > **Responses to the Follow-up Comments (1/2)**
> > > > >
> > > > > We much appreciate your follow-up comments to help further improve our paper. Here is our response to your questions:
> > > > >
> > > > > ---
> > > > >
> > > > > ***Q14: I believe there is a big misunderstand concerning what is the computational bottleneck in SAM. The authors seems to count how many variables they maintain, but what matter computationally is the number of backpropagations. EG-SAM and OG-SAM seems to have the same number of backpropagations per iterations, in which case the motivation of OG-SAM remains unclear to me.***
> > > > >
> > > > > We agree that the number of back-propagations are the same. However, note that the motivation for introducing OG-SAM is not only for reducing back-propagations. As discussed in Sec. 3.4, OG-SAM uniquely incorporates the past gradient $g_{t-1}$ at epoch $t$.
> > > > > This utilization of past information $g_{t-1}$ helps improve the performance. The concept of OG leveraging past gradients to improve performance is also discussed in (Rakhlin \& Sridharan, 2013)  and [4].
> > > > >
> > > > > ---
> > > > > ***Q15: For the (new) increasing $\rho$
> > > > >  baseline, it seem like the choice of $\rho$
> > > > >  is different from the other methods? The intention was to see whether increasing
> > > > >  (which can become identical to EG-SAM appropriate stepsize schedule) can perform similarly, so you need to at least pick $\rho$
> > > > >  similarly.***
> > > > >
> > > > >
> > > > > In the SAM increasing $\rho$ experiment, the final $\rho$ value $\rho_T = 1$, which aligns with
> > > > > $\rho$ value used in EG-SAM for CIFAR-10.
> > > > > The purpose of the experiment was to verify whether the incrementally increasing $\rho$ from a smaller number to the EG-SAM's $\rho$ value, mirroring EG-SAM's stepsize schedule, could enhance performance.
> > > > >
> > > > >
> > > > > Following the reviewer's feedback, we also conducted two additional experiments with SAM:
> > > > >
> > > > > 1) increasing
> > > > > $\rho$ from 0.05 (the value of $\rho$ in original SAM) to 1 (the value of $\rho$ in EG-SAM).
> > > > >
> > > > >
> > > > > 2) increasing $\rho$ from 0.90 ($\rho-\eta_0$, as $\rho=1.0$ for EG-SAM and $\eta_0=0.1$ as the initialized learning rate) to 1 (the value of $\rho$ in EG-SAM).
> > > > >
> > > > > Our results reveal that these increasing  $\rho$ methods all do not improve performance in various cases.
> > > > >
> > > > >
> > > > > We guess the reviewer also want to know further about our relationship to increased $\rho$ SAM. In EG-SAM, $\epsilon^\prime_t$ can be written as $\epsilon^\prime_t = \left(\frac{\rho - \eta_t ||\nabla L(w_{t-1})||}{||\nabla L(w_{t-1})||}\right) \nabla L(w_{t-1})$. Thus, for increasing \(\rho^\prime_t\) to align, it should satisfy $\frac{\rho - \eta_t ||\nabla L(w_{t-1})||}{||\nabla L(w_{t-1})||} = \frac{\rho^\prime_t}{||\nabla L(w_{t-1})||}$, leading to $\rho^\prime_t = \rho - \eta_t ||\nabla L(w_{t-1})||$. This explains why other methods with linearly increasing
> > > > > $\rho$ in SAM cannot perform worse than our method: our method takes the landscape information (the norm of the current gradient) into consideration when suppressing the perturbation.
> > > > >
> > > > >
> > > > > |                              | CIFAR-10 with ResNet-18     |
> > > > > |------------------------------|-----------------------------|
> > > > > | SAM (Foret et al., 2021)     | 96.52 ± 0.12                |
> > > > > | SAM increasing $\rho$ (0.1 to 1)     | 94.67 ± 0.03        |
> > > > > | SAM increasing $\rho$ (0.05 to 1)    | 94.84 ± 0.12        |
> > > > > | SAM increasing $\rho$ (0.9 to 1)    | 89.13 ± 0.23        |
> > > > > | EG-SAM                               | 96.86 ± 0.01        |
> > > > >
> > > > > ---
> > > > > ***Q16: You do implicitly assume function Lipschitz by assuming bounded variance.***
> > > > >
> > > > > Thank you for the reminder. We apologize that we missed the fact that bounded gradient/variance implies function Lipschitz.
> > > > >
> > > > >
> > > > > However, this does not alter our response to your original concern “the simultaneous requirement of both gradient-Lipschitz and function Lipschitz conditions being quite restrictive". As discussed in response to Q5, the assumption of both gradient-Lipschitz  and function Lipschitz (bounded gradient/variance) conditions is widely adopted in the analysis of normalized SAM and its variants (Mi et al., 2022; Zhang
> > > > > et al., 2023; Yue et al., 2023) and [1].
> > > > > This wide adoption in the field suggests that these assumptions are valid in the analysis of normalized SAM.

---

> > > > > > ### Author Response · Authors · 2023-11-22
> > > > > > **Responses to the Follow-up Comments (2/2)**
> > > > > >
> > > > > > (Continued)
> > > > > >
> > > > > > ---
> > > > > > ***Q17: I seems that the authors are misunderstanding my concern. The rates that the authors mentions concerning EG do not apply to their problem (which is convex in the max-player and not (strongly)-concave as otherwise assumed) and vice versa. I would be more careful with drawing parallels to classical minimax optimization as this can be misleading.***
> > > > > >
> > > > > > We appreciate your suggestion, and we firstly want to seek clarification on the statement "... which is convex in the max-player and not (strongly)-concave as otherwise assumed." Is this statement referred to the EG setting or our SAM problem setting? Our understanding is that for EG methods, the typical problem setting is non-convex and strongly concave, as described in (Mahdavinia et al., 2022). If this statement refers to our problem, as discussed in our response to Q6, it is a non-convex, non-concave setting. Both of these settings seems be unrelated to being "convex in the max-player."
> > > > > >
> > > > > >
> > > > > > We guess that your concern may be about the current EG's convergence rate $\mathcal{O}(T^{-1 / 4})$
> > > > > > is for non-convex strongly concave setting, which is a more stringent assumption compared to the non-convex, non-concave setting for SAM's problem.
> > > > > >
> > > > > >  However, note that without strongly concave assumption, the convergence rate is likely to be slower due to the lack of certain beneficial properties associated with strongly concave settings.
> > > > > > Therefore, the current convergence rate
> > > > > > for EG represents a specific case that shows the lower bound of the convergence rate under general non-convex non-concave setting.
> > > > > > We use this to illustrate the fact that applying EG/OG directly to SAM results slow convergence rate.
> > > > > >
> > > > > > We thank the reviewer again for reminding us to be more careful. we have incorporated the above analysis into Section 3.1 to enhance our paper's rigor.
> > > > > >
> > > > > > ---
> > > > > > ***Q18: What is the choice of
> > > > > >  of your method? You seems to search over a much more refined grid (0.01, 0.05, 0.08, 0.1, 0.5, 0.8, 1, 1.5, 1.8, 2), possibly explaining the difference in performance.***
> > > > > >
> > > > > > As discussed in Sec 5.1, we conduct grid search in
> > > > > > CIFAR-10, and we directly use it into other backbones and datasets.    The chosen $\rho=1.0$.
> > > > > >
> > > > > > Enlarging or decreasing $\rho$ from $1.0$ lead to worse performance. Specifically, enlarging $\rho$ results in worse performance, and when $\rho$ is large than 1.8, the accuracy drops less than 90\%. On the other hand,
> > > > > > decreasing the $\rho$ also decreases the performance,  but not as much.  The smallest value of $\rho$ we tested, $0.01$ has an accuracy around 95\%.
> > > > > >
> > > > > >
> > > > > >
> > > > > > ---
> > > > > > Reference:
> > > > > >
> > > > > > [4] Azizian, Waïss, et al. "The last-iterate convergence rate of optimistic mirror descent in stochastic variational inequalities." COLT 2021.

---

### Official Review · Reviewer_rtXV · 2023-11-01

**Soundness:** 3 good
**Presentation:** 3 good
**Contribution:** 2 fair
**Rating:** 6
**Confidence:** 3

**Summary:**

The paper tries to incorporate look-ahead mechanism into SAM to improve the performance of SAM. The authors proposed 3 methods: SAM + EG, SAM + OG and AO + SAM, in which the first one used extra-gradient information, the second method proposed to use optimistic-gradient and the last one used an adaptive gradient. The authors provided convergence analysis of those methods by proving that their convergence rates are similar to that of SAM. They also did some empirical works to verify the convergence rate, accuracies of their methods in CIFAR-10, CIFAR-100 and ImageNet using several network structures like ResNet-32, Resnet-18, WideResNet28-10.

**Strengths:**

The paper is well presented, clear structure.
The authors provides theoretical analysis for their methods. I did not check all the technical details, but I have a random check, it seems to be fine.
The experimental results have shown the improvement of the method over SAM. Even though the improvement is only around $1\%$, but it is shown consistently over several network structures.

**Weaknesses:**

Those proposed methods are only some variations of SAM with small changes.
Hence, the authors did not encounter further difficulties to obtain those convergence rate results.
The experiment results show  incremental enhancement  to SAM.
In comparison with SAM paper, the authors had done less experiments for comparisons, i.e.  ResNet-101, ResNet-152 etc
There is also a little intuitive explanation/proof for advantages of the proposed methods.
It is decent work, not something fancy to read.

**Questions:**

No question

---

> ### Author Response · Authors · 2023-11-20
> **Responses to Reviewer rtXV**
>
> We much appreciate your positive and insightful comments on our paper. Here is our response to your questions point by point:
>
> ---
>
> ***Q1: Those proposed methods are only some variations of SAM with small changes. Hence, the authors did not encounter further difficulties to obtain those convergence rate results.***
>
> i) Our proposed methods include the following contributions:
> 1) we improve SAM by using look-ahead mechanism to avoid saddle points.
> 2) We design different variants of the look-ahead mechanism, e.g., EG-SAM, OG-SAM, and AO-SAM, to further reduce computational cost.
> 3) We provide theoretical analysis.
> Experiments in Sec. 5 reveal that these modifications all help improve the performance of SAM.
>
>
>
> ii) Adding a look-ahead mechanism introduces new difficulties to the proof:
>
>
> 1) Recall in  OG-SAM, we have $\hat{w_t}=w_{t-1}-\eta_t g_{t-1}$, which includes past gradient $g_{t-1}$. This makes the standard SAM analysis (Andriushchenko \& Flammarion, 2022) difficult to apply directly, as their analytical approach does not include a technique to address past gradients.
>
> 2) In the proof of AO-SAM, we use an adaptive policy (Jiang et al., 2023) with OG. Although (Jiang et al., 2023) provides a proof, it is only established for unnormalized SAM. Since ours is normalized SAM with OG, it inherits the difficulties of both the adaptive policy and past gradient analysis.
>
> ---
> ***Q2: The experiment results show incremental enhancement to SAM.***
>
> i) Regarding the experimental results, while our method does not exhibit a
> dramatic improvement over the SOTA baselines, it consistently outperforms them
> across various environments and with different backbone models. Additionally, the
> baselines represent the current best-performing methods in the field. Therefore, it is common for the observed improvements to be relatively modest (Jiang et al.,
> 2023; Du et al., 2022a; Mi et al., 2022).
>
> ii) To further demonstrate that our improvement is NOT merely incremental, we have added a test of significance (T-test) on the noise-labeled CIFAR-100 dataset (Table 6). Our method shows statistical significance with a p-value less than 0.05. This supports our claim that our improvement is statistically significant, particularly for the noise-labeled dataset, which requires a higher degree of generalization ability.
>
> ---
>
> ***Q3: In comparison with SAM paper, the authors had done less experiments for comparisons, i.e. ResNet-101, ResNet-152 etc.***
>
> Due to resource limitations, we are sorry that our experiments did not include ResNet-101 and ResNet-152 on ImageNet.
>
> As an alternative, we incorporated an extra experiment with a large neural network, Vision Transformer ViT-S16 [1], on the CIFAR-10 and CIFAR-100 dataset. The results from this experiment demonstrate that our method  outperform other baselines continually.
>
>
>
> |       | CIFAR-10 with ViT-S16 | CIFAR-100 with ViT-S16 |
> |-------|-----------------------|------------------------|
> | SAM (Foret et al., 2021)  | 87.37 ± 0.09          | 63.23 ± 0.25           |
> | ESAM (Du et al., 2022a) | 84.27 ± 0.11          | 62.11 ± 0.15           |
> | OG-SAM| 88.27 ± 0.12          | 64.45 ± 0.23           |
>
>
> ---
>
> ***Q4: There is also a little intuitive explanation/proof for advantages of the proposed methods. It is decent work, not something fancy to read.***
>
> i) As discussed in Section 1, the introduction of the extra gradient in our model allows for gathering more information about the landscape by looking further ahead. This facilitates finding a better trajectory for convergence, which is the motivation for incorporating the extra-gradient into SAM.
>
>
> ii) Additionally, as also pointed out by Reviewer hY3b, the EG-SAM update scheme can be written as
> $w_t =w_{t-1}-\eta_t \nabla L\left( w_{t-1}+\epsilon^\prime_t\right)$ where
> $\epsilon^\prime_t = (\rho /||\nabla L(w_{t-1})||-\eta_t)\nabla L(w_{t-1})$.
> In comparison, SAM's update scheme is
> $w_t =w_{t-1}-\eta_t \nabla L\left( w_{t-1}+\epsilon_t\right)$, where
> $\epsilon_t = (\rho /||\nabla L(w_{t-1})||)\nabla L(w_{t-1})$. EG-SAM can thus be interpreted as it suppresses perturbation at the early stage (as $\epsilon_t$ decreases from large to small values), and allows more perturbation in the final stage. Since larger perturbations make the model more prone to being trapped in saddle points (Kim et al., 2023), our method more effectively avoids these points compared to SAM by reducing perturbation. Furthermore, the increased perturbation in the final stage enables the model to find flat minima.
>
> ---
>
> References:
>
> [1] Dosovitskiy, Alexey, et al. "An Image is Worth 16x16 Words: Transformers for Image Recognition at Scale." ICLR. 2020.

---

> > ### Comment · Reviewer_rtXV · 2023-11-23
> > **Responses to the rebuttal**
> >
> > I would like to thank the authors for their responses. I keep my initial score.

---

### Author Response · Authors · 2023-11-20
**Responses to all reviewers**

We first thank the reviewers for their valuable, constructive, and perspective comments. In response to your suggestions,  we have updated a revision of our paper, and here is a summary of our updates:


i) We have added a proof to allow the extrapolation step size $\eta$ to vary with respect to time $\tau$ in Sec. A.1.


ii) We have included a proof demonstrating that distance between the convergence stationary point found by EG-SAM and the stationary point of SAM's minimax problem (Eq. (1)) is bounded in Sec. A.5.



iii) We have added new experiments in Section B regarding:

1) OG-SAM with vision transformers on CIFAR-10 and CIFAR-100.

2) Gradient Norm Plot of SAM and EG-SAM.

3) Performance of SAM and EG-SAM with increasing $\rho$.

4) GSAM and MSAM baselines in CIFAR-10 and CIFAR-100.

5) The computational time of our methods and SAM.

We sincerely hope these revisions meet your expectations. More discussions and suggestions on further improving the paper are also always welcomed!

Thanks again!

---

### Meta-Review · Area_Chair_rLcT · 2023-12-07

**Metareview:**

The paper proposes modified variants of sharpness-aware minimization (SAM), a recent method to improve generalization in deep learning. The paper argues that SAM can get stuck in saddle-points, and the proposed variants (SAM+EG, SAM+OG and AO+SAM) avoid such saddle-points and therefore should work better. In numerics, some improvements of the proposed variants are shown.

The paper is easy to follow, and provides a good literature overview.  There are several weaknesses, the main one being that the proposed approach seems to be incremental, as extragradient variants or similar variants of SAM have been considered before and it is not clear what the paper adds to the literature. On the theoretical side there are strong assumptions in the proofs and it remains unclear whether how the theory relates to the presented numerical experiments.  In experiments, only a marginal improvement over other SAM techniques is shown and are limited to specific computer vision settings.

I encourage a resubmission, with a stronger focus on the underlying mechanisms of the proposed algorithms: Why should they work better? The main saddle-point hypothesis seems rather unlikely, given the highly stochastic and high-dimensional nature of the problem. Perhaps the proposed method can have benefits due to other reasons.

In summary, my recommendation is to reject this paper, but encourage the authors to take the reviewers comments into account for a resubmission.

**Justification For Why Not Higher Score:**

At the current stage, there are several weaknesses regarding the main claims of the submission (does the method really work better due to saddle-point avoidance) and issues with empirical/theoretical/novelty as pointed out by the reviewers.  Therefore I cannot recommend acceptance at this stage.

**Justification For Why Not Lower Score:**

N/A

---

### Decision · Program_Chairs · 2024-01-16

Reject